# Understanding the Role of Layer Normalization in Label-Skewed Federated Learning

**Guojun Zhang**                                    *guojun.zhang@huawei.com*
*Huawei Noah's Ark Lab*

**Mahdi Beitollahi**                                *mahdi.beitollahi@huawei.com*
*Huawei Noah's Ark Lab*

**Alex Bie**                                        *alexbie@huawei.com*
*Huawei Noah's Ark Lab*

**Xi Chen**                                         *xi.chen4@huawei.com*
*Huawei Noah's Ark Lab*

**Reviewed on OpenReview:** *https://openreview.net/forum?id=6BDHUkSPna*

## Abstract

Layer normalization (LN) is a widely adopted deep learning technique especially in the era of foundation models. Recently, LN has been shown to be surprisingly effective in federated learning (FL) with non-i.i.d. data. However, exactly why and how it works remains mysterious. In this work, we reveal the profound connection between layer normalization and the label shift problem in federated learning. To understand layer normalization better in FL, we identify the key contributing mechanism of normalization methods in FL, called *feature normalization* (FN), which applies normalization to the latent feature representation before the classifier head. Although LN and FN do not improve expressive power, they control feature collapse and local overfitting to heavily skewed datasets, and thus accelerates global training. Empirically, we show that normalization leads to drastic improvements on standard benchmarks under extreme label shift. Moreover, we conduct extensive ablation studies to understand the critical factors of layer normalization in FL. Our results verify that FN is an essential ingredient inside LN to significantly improve the convergence of FL while remaining robust to learning rate choices, especially under extreme label shift where each client has access to few classes. Our code is available at https://github.com/huawei-noah/Federated-Learning/tree/main/Layer_Normalization.

## 1 Introduction

Federated learning (FL, McMahan et al. 2017) is a privacy-enhancing distributed machine learning approach that allows clients to collaboratively train a model without exchanging raw data. A key challenge in FL is to handle data heterogeneity, where clients may have differing training data.

*Label shift* is a crucial data heterogeneity scenario where individual clients have varied label distributions. For instance: client 1 is a cat lover and only has cat images, while client 2 has a lot of dog images and few cat images. Such difference in client label distributions can cause substantial disagreement between the local optima of clients

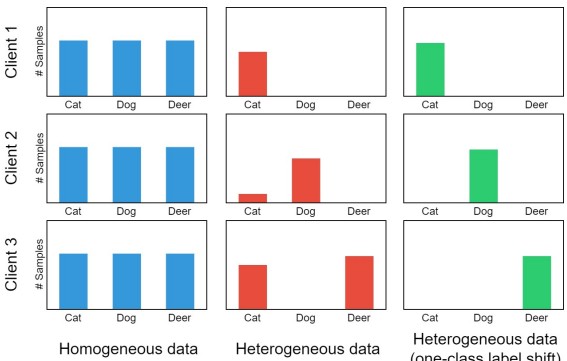

Figure 1: Visualization of label shift.

Table 1: Comparing normalization methods with state-of-the-art federated learning approaches. $n$ class(es) means that each client has only access to data from $n$ class(es). $\text{Dir}(\beta)$ denotes partitioning with symmetric Dirichlet distribution (Hsu et al., 2019). Our setup includes 10 clients for the CIFAR-10 dataset, 50 and 20 clients for the CIFAR-100 dataset with $n$ class(es) and $\text{Dir}(0.1)$, respectively, and 200 clients for TinyImageNet. Normalization (FedLN & FedFN) beats alternatives in a wide variety of settings; see Section 3 for further details.

| Methods | CIFAR-10 | | | CIFAR-100 | | | TinyImageNet | | |
|---|---|---|---|---|---|---|---|---|---|
| | 1 class | 2 classes | Dir(0.1) | 2 classes | 5 classes | Dir(0.1) | Dir(0.01) | Dir(0.02) | Dir(0.05) |
| FedAvg | 55.0 | 65.6 | 71.7 | 32.3 | 36.2 | 38.1 | 21.2 | 20.9 | 21.9 |
| FedProx | 54.7 | 66.7 | 71.4 | 32.4 | 36.6 | 38.8 | 21.0 | 20.8 | 22.1 |
| SCAFFOLD | 55.3 | 67.1 | 71.1 | 32.3 | 36.6 | 38.7 | 20.7 | 20.7 | 21.7 |
| FedLC | 10.0 | 57.9 | 65.0 | 6.0 | 18.2 | 22.3 | 0.8 | 0.4 | 0.5 |
| FedDecorr | 43.9 | 70.2 | 69.9 | 29.2 | 33.0 | 34.2 | 22.2 | 22.4 | 1.5 |
| FedRS | 10.0 | 56.7 | 66.1 | 10.7 | 24.9 | 22.1 | 14.8 | 18.4 | 19.0 |
| FedYogi | 80.9 | 80.0 | 79.2 | 44.6 | 44.1 | 45.2 | 22.5 | 24.3 | 25.7 |
| FedLN | **87.2** | **88.0** | **89.1** | **46.6** | **46.2** | **47.3** | **38.2** | **39.1** | **38.1** |
| FedFN | 80.8 | 82.3 | 84.2 | 40.3 | 45.4 | 45.4 | 34.0 | 34.1 | 34.1 |

and the desired global optimum, which negatively impacts FL performance. Taking label shift to the limit, a client may only have access to only *one class* data, as illustrated in Figure 1. This is the case for problems such as speaker identification, where each user has only their own voice samples. Our study focuses on such *extreme label shift*, where baseline FL approaches see drastic drops in utility.

Various methods have been proposed to mitigate the data heterogeneity problem in FL. For example, FedProx (Li et al., 2020a) adds $\ell_2$ regularization to avoid client model divergence. SCAFFOLD (Karimireddy et al., 2020) applies variance reduction to control the variance of client models. These frameworks focus on regularizing client drift in the FL process. FedLC (Zhang et al., 2022) and FedRS (Li & Zhan, 2021) attempt to address label shift by adjusting clients' model logits. We find that these techniques do not suffice to obtain good performance in the extreme label shift setting (see Table 1).

An orthogonal approach to the aforementioned methods is through normalizing the neural networks. In Hsieh et al. (2020), the authors point out the advantage of group normalization (Wu & He, 2018) in label-shifted FL problems over batch normalization (Ioffe & Szegedy, 2015). Du et al. (2022) attributes the shortcomings of batch normalization to external covariate shift. There has also been studies such as Li et al. (2020b) and Wang et al. (2023) that modify batch normalization for FL. Most closely related to our work is the very recent study of Casella et al. (2023) that experimentally compares various normalization methods in federated learning on MNIST and CIFAR-10.

**Our contributions.** Previous work lacks an in-depth analysis of layer normalization, and does not explain when and why layer normalization works in FL. In this work, we dive deeply into the theoretical and experimental analysis of layer normalization for federated learning. Our main finding is that *as client label distributions become more skewed, layer normalization helps more.* This is because under heavy label shift, each client easily overfits its local dataset without normalization, and LN effectively controls local overfitting by mitigating feature collapse. For example, in the extreme one-class setting (Figure 1), LN can improve over FedAvg by $\sim$ **32%** in terms of absolute test accuracy (see Table 1).

To further understand the effect of LN for FL, we ask the following question: *is there a much simplified mechanism that works equally well as LN?* The answer is positive. From our analysis of local overfitting and feature collapse, we discover that the key contributing mechanism of normalization methods in FL is *feature normalization* (FN), which applies normalization to the latent feature representation before the classifier head. Feature normalization simplifies LN while retaining similar performance (Table 1).

Another in-depth analysis we make is the comprehensive experiments and ablation studies. We compare LN and FN methods on a variety of popular datasets including CIFAR-10/100, TinyImageNet and PACS, with

various data heterogeneity and neural architectures such as CNN and ResNet. We are the *first* to show that layer normalization is the best method so far for label shift problems, outperforming existing algorithms like FedProx, SCAFFOLD and FedLC. Moreover, our ablation studies thoroughly analyze each factor of LN, including with/without mean-shift, running mean/variance, and before/after activation. Our empirical analysis is much deeper and broader than any of the previous results.

We summarize our contributions as follows:

- Under extreme label shift, we provide a comprehensive benchmark to clearly show the dramatic advantage of LN over popular FL algorithms for data heterogeneity;

- We are the first to propose and carefully analyze the suitability and properties of layer normalization in label-skewed FL problems, both theoretically and empirically;

- We discover the key mechanism of LN in such problems, feature normalization, which simply normalizes pre-classification layer feature embeddings.

## 2 Preliminaries and Related Work

In this section, we review the necessary background for our work, including federated learning and layer normalization.

**Federated learning (FL)** aims to minimize the objective $f(\boldsymbol{\theta}, \boldsymbol{W}) = \sum_{k=1}^{K} \frac{m_k}{m} f_k(\boldsymbol{\theta}, \boldsymbol{W})$, with each client loss as:

$$f_k(\boldsymbol{\theta}, \boldsymbol{W}) := \mathbb{E}_{(\boldsymbol{x}, y) \sim \mathcal{D}_k}[\ell(\boldsymbol{\theta}, \boldsymbol{W}; \boldsymbol{x}, y)], \tag{1}$$

Here $m_k$ is the number of samples of client $k$, and $m = \sum_k m_k$. We denote $\mathcal{D}_k$ as the data distribution of client $k$ and we have $m_k$ i.i.d. samples from it to estimate $f_k$. We use $\boldsymbol{\theta}$ to represent the parameter collection of the feature embedding network $\boldsymbol{g}_{\boldsymbol{\theta}}$ and $\boldsymbol{W} = (\boldsymbol{w}_1, \dots, \boldsymbol{w}_C)$ as the softmax weights for $C$-class classification. The per-sample loss is thus:

$$\ell(\boldsymbol{\theta}, \boldsymbol{W}; \boldsymbol{x}, y) = -\log \frac{\exp(\boldsymbol{w}_y^\top \boldsymbol{g}_{\boldsymbol{\theta}}(\boldsymbol{x}))}{\sum_{c \in [C]} \exp(\boldsymbol{w}_c^\top \boldsymbol{g}_{\boldsymbol{\theta}}(\boldsymbol{x}))}. \tag{2}$$

In the classical FL protocol (FedAvg, McMahan et al., 2017), a central server distributes the current global model to a subset of participating clients. During each communication round, selected clients perform multiple steps of gradient update on the received model with its local data and upload the updated model to the server. Finally, the server aggregates the clients' models to refine the global model.

**Class imbalance problem in FL.** Since its inception, federated learning studies the class imbalance problem, i.e., different clients have different label distributions. In the seminal FL paper (McMahan et al., 2017), the authors considered partitioning the dataset with different shards, and giving each client two shards. In this way, each client would have two class labels. Another common way to create such class imbalance is to use Dirichlet distribution (Hsu et al., 2019). FedAwS (Yu et al., 2020) considered the extreme case when we have one class per each client. FedProx (Li et al., 2020a), FedLC (Zhang et al., 2022) and SCAFFOLD (Karimireddy et al., 2020) addressed class imbalance by modifying the objective or the model aggregation. FedOpt (Reddi et al., 2020) considered adaptive aggregation as inspired by adaptive optimization such as Adam (Kingma & Ba, 2014). More recently, FedDecorr (Shi et al., 2023) pointed out the feature collapse in federated learning and proposes a new algorithm to mitigate data heterogeneity. Shen et al. (2022) proposed CLIMB which studies class imbalance FL problems using constrained optimization.

**Normalization.** Suppose $\boldsymbol{x} \in \mathbb{R}^d$ is a vector, we define the *mean-variance* (MV) normalization of $\boldsymbol{x}$ as:

$$\mathsf{n}_{\mathsf{MV}}(\boldsymbol{x}) := \frac{\boldsymbol{x} - \mu(\boldsymbol{x})\mathbf{1}}{\sigma(\boldsymbol{x})}, \ \mu(\boldsymbol{x}) = \frac{1}{d} \sum_{i=1}^{d} x_i, \ \sigma(\boldsymbol{x}) = \sqrt{\frac{1}{d} \sum_{i=1}^{d} (x_i - \mu(\boldsymbol{x}))^2}. \tag{3}$$

Here $\mu$ and $\sigma$ are standard notions of mean and variance. Symbolically, a layer normalized feed-forward neural net is a function of the following type:

$$\mathsf{n}_{\mathsf{MV}} \circ \rho \circ \boldsymbol{A}_L \circ \mathsf{n}_{\mathsf{MV}} \circ \rho \circ \boldsymbol{A}_{L-1} \dots \mathsf{n}_{\mathsf{MV}} \circ \rho \circ \boldsymbol{A}_1, \tag{4}$$

where $\rho$ is the activation and $\boldsymbol{A}_i(\boldsymbol{a}) = \boldsymbol{U}_i \boldsymbol{a} + \boldsymbol{b}_i$ is an affine function for a vector $\boldsymbol{a}$.

We can generalize MV normalization to tensors by considering the mean and variance along each dimension. Based on the discussion above, *batch normalization* (BN, Ioffe & Szegedy 2015) is just MV normalization along the direction of samples, and *layer normalization* (LN, Ba et al. 2016) is MV along the direction of hidden neurons. Unlike BN, layer normalization can be applied to batches of any size and does not require statistical information of batches of each client. Other normalization methods include group normalization (Wu & He, 2018), weight normalization (Salimans & Kingma, 2016) and instance normalization (Ulyanov et al., 2016). Other than normalization on the neural network, Francazi et al. (2023) proposed normalizing the per-class gradients to address the class imbalance problem.

MV normalization can be rewritten as:

$$\mathsf{n}_{\mathsf{MV}}(\boldsymbol{x}) = \sqrt{d}\frac{\boldsymbol{x} - \mu(\boldsymbol{x})\mathbf{1}}{\|\boldsymbol{x} - \mu(\boldsymbol{x})\mathbf{1}\|} = \sqrt{d} \cdot \mathsf{n}'(\boldsymbol{x} - \mu(\boldsymbol{x})\mathbf{1}), \tag{5}$$

with $\mathsf{n}'(\boldsymbol{x}) = \boldsymbol{x}/\|\boldsymbol{x}\|$. In other words, it is a composition of mean shift, division by its norm, and a scaling operation with factor $\sqrt{d}$ (see also Brody et al., 2023). If $\mu(\boldsymbol{x}) = \mathbf{0}$, we obtain

$$\mathsf{n}(\boldsymbol{x}) = \sqrt{d} \cdot \mathsf{n}'(\boldsymbol{x}).$$

We call the function $\mathsf{n}$ as the *scale normalization*, which shares similarity with RMSNorm (Zhang & Sennrich, 2019). This function retains scale invariance but loses shift invariance. To improve stability, we can replace $\boldsymbol{x}/\|\boldsymbol{x}\|$ with $\boldsymbol{x}/\max\{\epsilon, \|\boldsymbol{x}\|\}$.

We may similarly construct another normalized neural network by replacing $\mathsf{n}_{\mathsf{MV}}$ with $\mathsf{n}$ in eq. 6:

$$\mathsf{n} \circ \rho \circ \boldsymbol{A}_L \circ \mathsf{n} \circ \rho \circ \boldsymbol{A}_{L-1} \cdots \circ \mathsf{n} \circ \rho \circ \boldsymbol{A}_1, \tag{6}$$

which we will call a *feature normalized (FN)* neural net. This name will be clear with Proposition 1.

**Normalization in FL.** The fact that group norm (Wu & He, 2018) is better than batch norm in FL was observed in Hsieh et al. (2020). Li et al. (2020b) proposed FedBN and adapted batch normalization in FL, where each client model is personalized. Analysis of batch norm in FL can be found in Du et al. (2022) and Wang et al. (2023). In Casella et al. (2023), the authors tested different normalization methods in FL on non-i.i.d. settings. Compared to the these papers, we are the first to observe and analyze the connection between layer/feature normalization and label shift.

## 3   Efficacy of Layer Normalization

**Experimental setting.** To show the efficacy of layer normalization in label-skewed FL problems, we conduct an extensive benchmark experiment to compare FedAvg + LN (FedLN) and FedAvg + FN (FedFN) with several popular FL algorithms, including the original FedAvg (McMahan et al., 2017), in addition to:

- Regularization based methods: FedProx (Li et al., 2020a), SCAFFOLD (Karimireddy et al., 2020) and FedDecorr (Shi et al., 2023);
- Logit calibration methods for label shift: FedLC (Zhang et al., 2022) and FedRS (Li & Zhan, 2021);
- Adaptive optimization: FedYogi (Reddi et al., 2020).

To simulate the label shift problem, we create two types of data partitioning. One is $n$ class(es) partitioning where each client has only access to data from $n$ class(es). Another is Dirichlet partitioning from Hsu et al. (2019). As discussed in Hsieh et al. (2020), the label skew problem is pervasive and challenging for decentralized training.

We test the comparison on several common datasets including CIFAR-10, CIFAR-100 (Krizhevsky et al., 2009) and TinyImageNet (Le & Yang, 2015) with CNN and ResNet-18 (He et al., 2016).

**Results.** Our results are displayed in Table 1. We conclude that:

- Under extreme label shift, all baseline algorithms do not show a clear edge over the vanilla FedAvg algorithm, except FedYogi. This demonstrates the frustrating challenge of label skewness.
- FedYogi can drastically improve FedAvg in such cases (e.g. CIFAR-10), but on larger datasets like TinyImageNet, the improvement is marginal.
- FedLN is the only algorithm that can dramatically improve FedAvg in all scenarios.
- FedFN largely captures the performance gain of FedLN, despite being consistently inferior to FedLN.

Because of the drastic improvement of layer norm and feature norm in label-skewed problems, we will carefully explore the reasons behind, both theoretically and experimentally, in the following sections.

## 4 Why Is Layer Normalization So Helpful?

Inspired by the success of layer/feature norms, we wish to obtain some primary understanding of the reason why they are so suitable under label shift. Our main findings include: (**a**) Layer norm and feature norm can be reduced to the last-layer normalization; (**b**) the last-layer normalization helps address feature collapse which happens in the local overfitting of one-class datasets. This accelerates the training of feature embeddings. (**c**) the main advantages of LN/FN lie in training process, rather than the expressive power.

### 4.1 Delegating scaling to the last layer

We first show that both feature normalized and layer normalized neural networks can be simplified to the last-layer scaling, under the assumption of scale equivariance.

**Assumption 1.** *The bias terms from the second layer onward are all $\mathbf{0}$, i.e., $\boldsymbol{b}_i = \mathbf{0}$ for $i = 2, \ldots, L$, and the activation function is (Leaky) ReLU.*

For an input $\boldsymbol{x}$, denote the $i$th activation as $\boldsymbol{a}_i := \rho \circ \boldsymbol{A}_i \ldots \rho \circ \boldsymbol{A}_1(\boldsymbol{x})$. Under Assumption 1, a vanilla neural network is scale equivariant w.r.t. the first layer activation, i.e., suppose $\lambda > 0$ and $h := \rho \circ \boldsymbol{A}_L \ldots \rho \circ \boldsymbol{A}_2$ is the neural network function from the second layer then $h(\lambda \boldsymbol{a}_1) = \lambda h(\boldsymbol{a}_1)$ holds. Assumption 1 is also necessary for such equivariance.

**Proposition 1** (**reduced feature normalization**). *Under Assumption 1, scale normalizing each layer is equivalent to only scale normalizing the last layer. That is, for all affine transformations $\boldsymbol{A}_1, \ldots, \boldsymbol{A}_L$ the function*

$$\mathsf{n} \circ \rho \circ \boldsymbol{A}_L \circ \mathsf{n} \circ \rho \circ \boldsymbol{A}_{L-1} \ldots \mathsf{n} \circ \rho \circ \boldsymbol{A}_1$$

*is equal to*

$$\mathsf{n} \circ \rho \circ \boldsymbol{A}_L \circ \rho \circ \boldsymbol{A}_{L-1} \ldots \rho \circ \boldsymbol{A}_1 \tag{7}$$

*if all the intermediate hidden vectors after activation are non-zero.*

The formal statement of this proposition can be found in Proposition 1'. In this proposition, we used blue color to highlight the normalization components before and after the simplification. Details of Proposition 1 can be found in the appendices. Eq. 7 can be considered as scale normalizing the feature embedding with a vanilla network, which is why we called it feature normalization in eq. 6. Note that Proposition 1 tells us that starting from the same random initialization for $\boldsymbol{A}_1, \ldots, \boldsymbol{A}_L$, training both formats will result in the same final network.

Similarly, we can simplify a layer-normalized neural network, by defining the shift operator:

$$\mathsf{s}(\boldsymbol{x}) = \boldsymbol{x} - \mu(\boldsymbol{x})\mathbf{1},$$

where $\boldsymbol{x}$ is an arbitrary finite-dimensional vector.

**Proposition 2** (**reduced layer normalization**)**.** *Under Assumption 1, MV normalizing each layer is equivalent to only MV normalizing the last layer and shifting previous layers. That is, for all affine transformations $\boldsymbol{A}_1, \ldots, \boldsymbol{A}_L$ the function*

$$\mathsf{n}_{\mathsf{MV}} \circ \rho \circ \boldsymbol{A}_L \circ \mathsf{n}_{\mathsf{MV}} \circ \rho \circ \boldsymbol{A}_{L-1} \ldots \mathsf{n}_{\mathsf{MV}} \circ \rho \circ \boldsymbol{A}_1$$

*is equal to*

$$\mathsf{n}_{\mathsf{MV}} \circ \rho \circ \boldsymbol{A}_L \circ \mathsf{s} \circ \rho \circ \boldsymbol{A}_{L-1} \cdots \circ \mathsf{s} \circ \rho \circ \boldsymbol{A}_1 \tag{8}$$

*if none of the intermediate hidden layer activations to normalize are proportional to the all-one vector* **1**.

The formal statement of Proposition 2 can be found in Proposition 2'.

**Extension to ResNet.** So far we have talked about simple feed-forward networks (including MLP and CNN). We can extend our results of Prop. 1 and Prop. 2 to other model architectures like ResNet (He et al., 2016), as long as the original model is scale equivariant. Note that ResNet is a composition of multiple blocks, where each block is:

$$\mathsf{block}(\boldsymbol{x}) = \rho(\boldsymbol{x} + \boldsymbol{A}_2 \circ \rho \circ \boldsymbol{A}_1(\boldsymbol{x})), \tag{9}$$

where each of $\boldsymbol{A}_1, \boldsymbol{A}_2, \boldsymbol{A}_3$ is a linear transformation. For any $\lambda > 0$, we would have $\mathsf{block}(\lambda\boldsymbol{x}) = \lambda\mathsf{block}(\boldsymbol{x})$. If we add layer normalization to this block, it becomes:

$$\mathsf{block}_{\mathsf{LN}}(\boldsymbol{x}) = \mathsf{n}_{\mathsf{MV}} \circ \rho(\boldsymbol{x} + \boldsymbol{A}_2 \circ \mathsf{n}_{\mathsf{MV}} \circ \rho \circ \boldsymbol{A}_1(\boldsymbol{x})). \tag{10}$$

A similar conclusion as Prop. 2 would be to replace the block above as:

$$\mathsf{block}'_{\mathsf{LN}}(\boldsymbol{x}) = \mathsf{s} \circ \rho(\boldsymbol{x} + \boldsymbol{A}_2 \circ \mathsf{n}_{\mathsf{MV}} \circ \rho \circ \boldsymbol{A}_1(\boldsymbol{x})). \tag{11}$$

However, we cannot replace the second $\mathsf{n}_{\mathsf{MV}}$ with mean shift, which causes a larger gap between LN and only normalizing the last layer feature as we will see in Table 2.

Our conclusions do not immediately extend to Vision Transformers (Dosovitskiy et al., 2020), since the multi-head attention is not scale equivariant due to the use of softmax.

## 4.2 Expressive power

The reduction of feature norm and layer norm allows us to analyze the expressive power of LN/FN networks. Denote $\varepsilon$, $\varepsilon_F$, $\varepsilon_L$ as the 0-1 classification error of a vanilla/FN/LN network respectively. We have:

**Proposition 3** (**expressive power**)**.** *Given any model parameters $(\boldsymbol{\theta}, \boldsymbol{W})$ and any sample $(\boldsymbol{x}, y)$ with $\boldsymbol{g}_{\boldsymbol{\theta}}(\boldsymbol{x}) \neq \boldsymbol{0}$, vanilla and FN networks have the same error on sample $(\boldsymbol{x}, y)$, i.e., $\varepsilon(\boldsymbol{\theta}, \boldsymbol{W}; \boldsymbol{x}, y) = \varepsilon_F(\boldsymbol{\theta}, \boldsymbol{W}; \boldsymbol{x}, y)$. For any model parameter $(\boldsymbol{\theta}, \boldsymbol{W})$ of a layer normalized network, one can find $(\boldsymbol{\theta}', \boldsymbol{W}')$ of a vanilla model such that $\varepsilon(\boldsymbol{\theta}', \boldsymbol{W}'; \boldsymbol{x}, y) = \varepsilon_L(\boldsymbol{\theta}, \boldsymbol{W}; \boldsymbol{x}, y)$, for any $\boldsymbol{x}, y$.*

Let us explain the significance of this proposition. In this work we aim to understand the role of layer normalization in federated learning. There could be two reasons why LN is so helpful:

- Layer normalization is more powerful in that it can express better functions to fit the data.

- Layer normalization allows the training process to be faster.

Proposition 3 rules out the first explanation. It tells us that in principle, a vanilla network can learn any pattern that LN/FN networks generate. If an LN/FN network could reach 80% accuracy, so can a vanilla network by adapting its parameters. Moreover, the classes of FN and vanilla networks are equivalent, but FN still outperforms vanilla networks in our settings. This rules out the explanation of model class restriction. Therefore, the benefit of LN/FN we saw in Table 1 lies in the training process, which we will present in the next subsection.

There still remains an important question regarding Proposition 3: since vanilla NNs can express LN networks, is the reverse true? Or could LN networks express any "meaningful" NN functions that fit the data? This question is out of our scope and we leave it to future research.

Table 2: Comparing the difference between LN and FN for CNNs and ResNet. We observe a smaller accuracy gap for CNNs, as predicted by our theory.

| Methods | 1 class | 2 classes | Dir(0.1) |
|---|---|---|---|
| FedLN - CNN | 78.04 | 77.52 | 78.27 |
| FedFN - CNN | 77.14 | 76.67 | 78.13 |
| FedLN - ResNet | 86.15 | 88.01 | 89.06 |
| FedFN - ResNet | 80.81 | 82.30 | 84.19 |

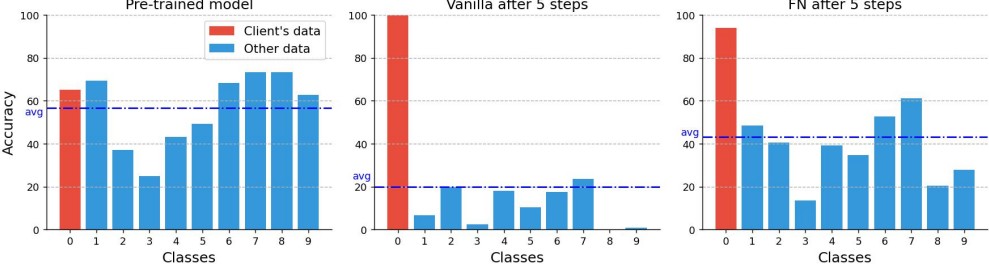

Figure 2: Local overfitting in the one-class setting on CIFAR-10. The client only has examples from class 0. The blue lines show the average global performance. (**left**) the test accuracies of the pre-trained model before local training; (**middle**) after 5 steps of local training with a vanilla model; (**right**) after 5 steps of local training with FN. Best viewed in color.

### 4.3 Normalization is essential for label shift

Now that we have simplified normalized networks to the last-layer scaling, our next question is: what is the relation between last-layer scaling and label shift? In order to further illustrate this connection, we consider the most extreme one-class setting, where each client has samples from only one class (see Figure 1). We defer the exploration of more general label shift to future work.

**Label shift induces local overfitting.** The scarcity of label variation on a client poses a risk of local overfitting, which we define below:

> In FL, *local overfitting* describes the situation when a client model performs extremely well on its local dataset, but fails to generalize to other clients.

We illustrate local overfitting in Figure 2, where the client dataset only has samples from class 0. After local training, the vanilla model easily reaches 100% on its local dataset, while the performance on other clients drastically drops. This is true even when it is initialized from a relatively good pre-trained model (left figure). This resembles the well-known phenomenon of *catastrophic forgetting* (McCloskey & Cohen, 1989). Comparably, feature normalization can mitigate local overfitting (Figure 2, right).

Let us try to understand local overfitting more clearly. Given the dataset $S_k = \{(\boldsymbol{x}_i, k)\}_{i=1}^{m_k}$ for client $k$, feature embedding $\boldsymbol{g_\theta}$ and class weight vectors (class embeddings) $\boldsymbol{w}_k$'s, the cross-entropy loss of client $k$, eq. 1 becomes:

$$f_k(\boldsymbol{\theta}, \boldsymbol{W}) = \mathbb{E}_{(\boldsymbol{x}, y) \in S_k} \left[ \log \sum_{c \in [C]} \exp((\boldsymbol{w}_c - \boldsymbol{w}_k)^\top \boldsymbol{g_\theta}(\boldsymbol{x})) \right],$$

Minimizing this loss function drives $(\boldsymbol{w}_c - \boldsymbol{w}_k)^\top \boldsymbol{g_\theta}(\boldsymbol{x}_i) \to -\infty$ for all $\boldsymbol{x}_i$ and $c \neq k$, leading to pathological behavior. For example, as long as $(\boldsymbol{w}_c - \boldsymbol{w}_k)^\top \boldsymbol{g_\theta}(\boldsymbol{x}_i) < 0$ for all $i$ and $c \neq k$, both training and test accuracies reach 100%, which often occurs in our practice. We present a necessary condition to minimize $f_k(\boldsymbol{\theta}, \boldsymbol{W})$:

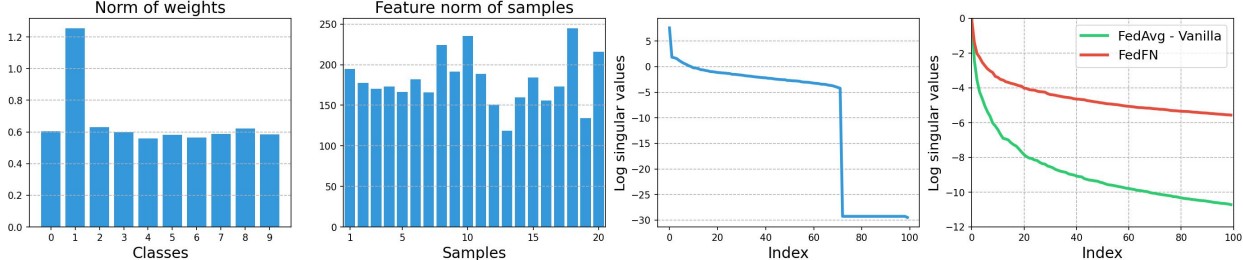

Figure 3: Local training with only samples from one class. **(left)**: vector norms of each class embedding. **(middle left)**: the norms of different feature vectors. We randomly choose 20 images from the dataset. **(middle right)**: singular values of features of a local overfitted model; **(right)**: singular values of normalized features learned from FedAvg and FedFN.

**Theorem 1** (**divergent norms**). *In order to minimize the one-class local loss $f_k(\boldsymbol{\theta}, \boldsymbol{W})$, we must have at least one of the following: 1) $\|\boldsymbol{w}_c\| \to \infty$ for all $c \in [C]$ and $c \neq k$; 2) $\|\boldsymbol{g}_{\boldsymbol{\theta}}(\boldsymbol{x}_i)\| \to \infty$ for all $\boldsymbol{x}_i$; 3) $\|\boldsymbol{w}_k\| \to \infty$.*

Theorem 1 tells us that either some class embedding norms or all feature embedding norms must diverge. Thus it points out the importance of controlling the feature/class embedding norms. If we do not add any feature/layer normalization, minimizing the label skewed local dataset could result in divergent norms.

**Which norms diverge?** In Figure 3, we perform local training on a client with samples from only one class of the standard CIFAR-10 dataset. In the left panel, we calculate the norms of each $\boldsymbol{w}_c$ with $c \in [10]$. It can be shown that $\|\boldsymbol{w}_1\|$ is relatively large compared to others. In the middle-left panel, we sample 20 images from client 1 and compute their feature norms. We see that $\|\boldsymbol{g}_{\boldsymbol{\theta}}\|$'s are all very large, compared to class embeddings. Combining with Theorem 1 and Figure 3, we can argue that controlling the divergence feature norms is more important, which emphasizes the importance of LN/FN in label-skewed FL.

**Feature collapse explains local overfitting.** If we plot the singular values of the feature matrix of 20 random images from different classes, $[\boldsymbol{g}_{\boldsymbol{\theta}}(\boldsymbol{x}_1), \ldots, \boldsymbol{g}_{\boldsymbol{\theta}}(\boldsymbol{x}_{20})]$, we observe a huge spectral gap. While the first singular value $\sigma_1 \approx 1.9 \times 10^3$, the second singular value is $\sigma_2 \approx 6.9$, which implies that the feature embeddings are approximately in a one-dimensional subspace. This is known as *feature collapse* (c.f. Shi et al. 2023), as the feature embeddings of different classes are mapped to the same directions. This explains the local overfitting of one-class datasets.

**Normalization addresses feature collapse.** In fact, in the one-class setting, there is no need for the feature embedding to distinguish images from different classes, especially when there is not much variation in the local dataset (such as the one-class setting). For example, the embedding can simply increase the feature norms without changing the directions.

In contrast, if the feature norms are constrained (like in LN/FN), then each client cannot overfit by increasing the feature norms, but has to learn the directional information of the feature embeddings. This accelerates the training of feature embeddings under heavy label shift.

We verify this claim on the right of Figure 3. While a vanilla network learns degenerate features, an FN network mitigates feature collapse and the feature embedding has more directions that are effective, measured by the corresponding singular values (c.f. Shi et al. 2023).

**Communication cost and privacy.** Adding FN or LN does not increase the number of parameters of the network, and only cause minor additional computation at each client. Therefore, vanilla FedAvg and FedFN have the same communication cost for sending models to a server at *each communication round*. Further, as implied in Table 1, due to faster convergence, FedFN requires less communication rounds to reach a pre-defined accuracy, and therefore, FedFN has a better total communication cost.

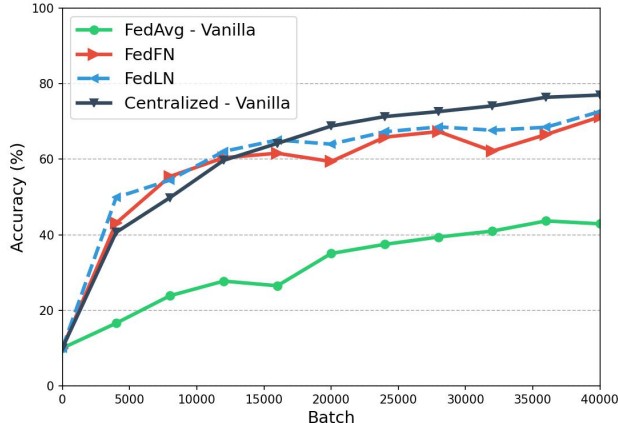

Figure 4: Comparing test accuracies of centralized training with FedAvg of models with different normalizations in one-class label shift in CIFAR-10. The $x$-axis denotes how much data is fed into an algorithm, measured by batches.

Table 3: Comparing layer-wise vs. last-layer normalization. $\mathsf{n}^L$ and $\mathsf{n}\mathbb{1}^{L-1}$ denote layer-wise and last-layer scaling respectively; they perform similarly as predicted by Prop. 1. $\mathsf{n}_{\mathsf{MV}}^L$ and $\mathsf{n}_{\mathsf{MV}}\mathsf{s}^{L-1}$ denote layer-wise and last-layer MV normalization respectively; Prop. 2 predicts they perform similarly. The table shows that the assumption on bias terms as required by Assumption 1 does not affect the performance. Our setup includes 10 clients.

| Methods | without bias | | | with bias | | |
|---|---|---|---|---|---|---|
| | 1 class | 2 classes | Dir(0.1) | 1 class | 2 classes | Dir(0.1) |
| FedAvg | 56.82 | 71.83 | 73.06 | 58.57 | 72.0 | 73.17 |
| FedFN - $\mathsf{n}^L$ | 77.05 | 76.60 | 77.27 | 77.70 | 76.29 | 76.50 |
| FedFN - $\mathsf{n}\mathbb{1}^{L-1}$ | 77.14 | 76.67 | 78.14 | 74.46 | 76.27 | 76.78 |
| FedLN - $\mathsf{n}_{\mathsf{MV}}^L$ | 77.16 | **78.09** | 78.71 | **77.75** | 76.73 | **78.36** |
| FedLN - $\mathsf{n}_{\mathsf{MV}}\mathsf{s}^{L-1}$ | **77.71** | 77.65 | **79.94** | 76.89 | **76.74** | 77.26 |

Unlike BN which requires the record of the running statistics of data during training, LN and FN do not record statistics of batches. Instead, in LN and FN, the normalization is done independently for each sample both in training and inference. Therefore, adding FN and LN to an FL system is unlikely to degrade privacy.

## 5 Experimental Analysis

We conduct extensive experiments to further analyze the relation between LN/FN and label-shifted federated learning.

### 5.1 FN is the essential mechanism of LN

In Figure 4, we compare different FedAvg algorithms with centralized training in the one-class setting of CIFAR-10. We apply different normalization to a CNN architecture.

If we apply vanilla unnormalized networks to FedAvg, the average test accuracy grows very slowly compared to centralized training, although the same amount of data is passed to both algorithms. Comparably, the convergence of FedAvg with LN and FN networks is much faster and closer to centralized training. Moreover, the performance of FedFN/FedLN are very similar to each other, confirming that FN is the main mechanism of LN under extreme label shift.

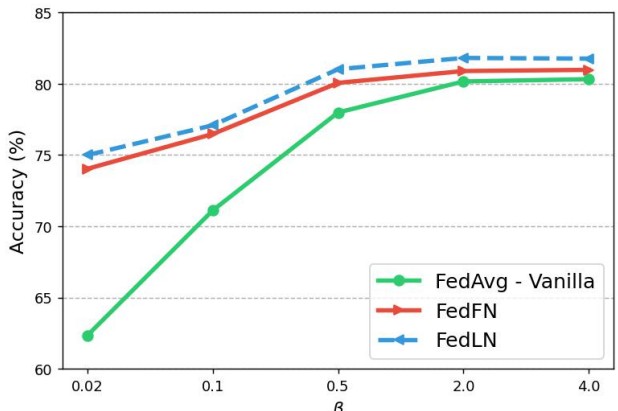

Figure 5: Effect of data heterogeneity on the performance of normalization. $\beta$ represents the parameter in the Dirichlet distribution that is used to sample client label distributions.

Table 4: Comparison among normalization methods in FL. FedAvg + BN means average all the parameters after using batch normalization. FedBN is from Li et al. (2020b). We use 10 clients with CIFAR-10 dataset.

| Methods | 1 class | 2 classes | Dir(0.1) |
|---|---|---|---|
| FedAvg | 56.82 | 71.83 | 73.06 |
| FedGN | 77.15 | 77.54 | 79.15 |
| FedAvg + BN | 9.63 | 24.64 | 58.80 |
| FedBN | 10.00 | 19.90 | 41.25 |
| FedFN | 77.14 | 76.67 | 78.14 |
| FedLN | **77.71** | **77.65** | **79.94** |

We also verify Prop. 1 and Prop. 2 by running FN/LN before and after simplification. To satisfy Assumption 1, we removed the bias terms in the neural net and kept the ReLU activation. Table 3 verifies that up to statistical error, normalization methods can be simplified to last-layer normalization. Even more, our simplification helps slightly as the last-layer normalization avoids division in the middle. In all cases, the performance of FN is very close to LN.

We also compare the performance gap between FN and LN with different architectures, including CNN and ResNet. From Table 2 we find that the gap with CNN is smaller than that of ResNet. This agrees with our discussion following eq. 11: the larger gap is due to the LN inside each block that cannot be absorbed into FN. We present additional experimental results regarding this point in the appendices.

## 5.2 Does normalization always help?

One might argue that layer norm is just advantageous in general and it has no relation with label shift. We show that is not the case. In Figure 5, we compare FedFN/FedLN with FedAvg with different levels of label skewness. Here $\beta$ controls the level of label skewness. When $\beta$ is small, the label shift is heavy and there is a clear gap between FedFN/FedLN and FedAvg. However, when the clients are more i.i.d., the performance gap diminishes. This reveals the strong connection between LN/FN and label shift.

## 5.3 Do other normalization methods help?

We compare FN/LN with other candidates of normalization, including group norm (GN, Wu & He, 2018) and batch norm (BN, Ioffe & Szegedy, 2015).

Table 4 shows the comparison in the one-class FL setting. For FedFN/FedLN we use Prop. 1 and Prop. 2 to simplify the normalization layers. We compare FedFN/FedLN with FedGN (FedAvg + GN,

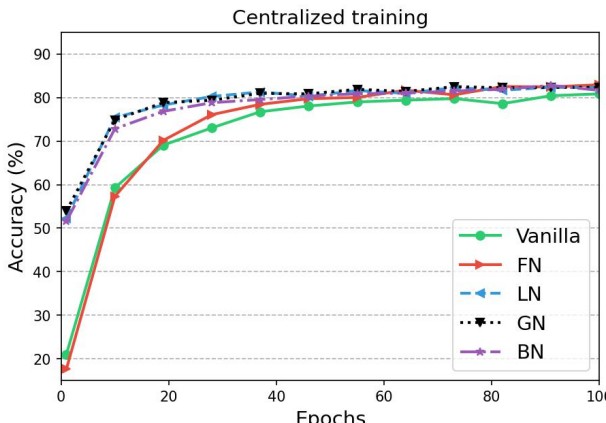

Figure 6: Comparison of different normalization methods under the centralized setting.

Table 5: PACS dataset comparison. 2-2-3 classes means we split each domain into 3 clients, with 2 classes, 2 classes and 3 classes respectively.

| Methods | 2-2-3 classes | Dir(0.5) | Dir(1.0) |
|---------|---------------|----------|----------|
| FedAvg  | 54.35         | 67.60    | 68.15    |
| FedProx | 55.96         | 57.14    | 56.58    |
| FedYogi | 64.56         | 71.81    | 72.20    |
| FedFN   | **71.58**     | **72.72**| 71.89    |
| FedLN   | 70.46         | 69.39    | **72.72**|

group_number=2) and BN. Note that there are two ways to apply batch normalization: one is simply adding BN as we did for FN/LN/GN, another is to avoid averaging the running mean/variance of the BN layers and keep them local, as suggested by Li et al. (2020b).

As argued in Du et al. (2022); Hsieh et al. (2020); Li et al. (2020b); Wang et al. (2023), plainly adding BN does not improve FedAvg. Our Table 4 verifies this conclusion and shows that FedAvg + BN could even worsen the performance, especially under heavy label shift. Moreover, each after personalizing the BN layers of each client (Li et al., 2020b), FedBN does not help mitigate label shift either. One explanation is that FedBN was designed for covariate shift problems, while our main setting is label shift.

On the other hand, FedGN behaves similarly to FedLN and FedFN. This is expected since group norm can be treated as a slight generalization of layer norm: it splits the hidden vector into subgroups and MV normalizes each group. If the group number is one, then GN reduces to LN.

However, the situation differs in the centralized setting, as we see in Figure 6. When we aggregate all the data on the server and perform different types of normalization, we find that BN/LN/GN improves centralized training similarly, as observed in recent literature (Wu & He, 2018). In contrast, the performance gain of FN disappears. This again verifies the strong connection between FN and label shift.

### 5.4 Label shift under covariate shift

In real scenarios, we may also face other challenges than label shift. For example, covariate shift may also be a great challenge in FL which occurs when the input distributions differ. We test different normalization methods under covariate shift using a classic dataset called PACS (Li et al., 2017), in addition to our main problem, label shift. The results can be seen in Table 5. From the table we conclude that FedFN/FedLN still has a clear edge under covariate shift.

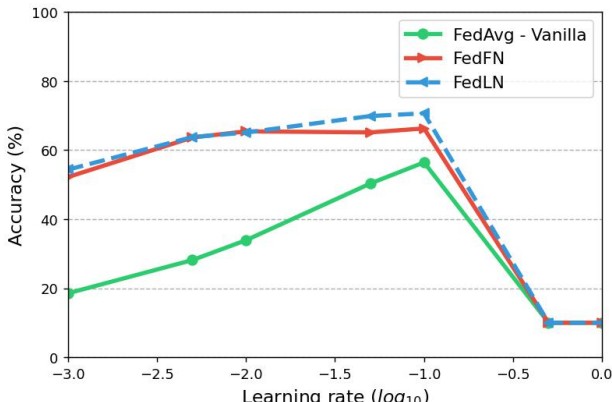

Figure 7: Effect of learning rate on the performance of FN, LN and vanilla networks in one-class distribution setting.

Table 6: Comparison among different variations of FedLN.

| Methods | 1 class | 2 classes | Dir(0.1) |
|---|---|---|---|
| FedLN - learnable | **78.04** | 77.52 | **78.27** |
| FedLN - static | 77.17 | **78.41** | **78.27** |
| FedLN - before | 77.04 | 78.08 | **78.27** |

### 5.5 Learning rate robustness

In Figure 7, we show the performance of FN, LN, and vanilla methods in the one-class setting of CIFAR-10. With different learning rates, the performance of the vanilla method changes severely, while the variation of FN/LN is relatively small within a large range of learning rates. This shows the ease of hyperparameter tuning with normalization.

### 5.6 Variation of layer normalization

In the implementation of LN, we have in fact implemented learnable parameters (FedLN - learnable), by adding an element-wise affine transformation to the vector:

$$\boldsymbol{\gamma} \odot \frac{\boldsymbol{x} - \mu(\boldsymbol{x})\mathbf{1}}{\sigma(\boldsymbol{x})} + \boldsymbol{\beta}.$$

If we fix $\boldsymbol{\gamma} = \mathbf{1}$ and $\boldsymbol{\beta} = \mathbf{0}$, then it reduces to a usual MV normalization, which we call the *static* setting. We compare both methods in Table 6, where we also add using LN before the activation (FedLN - before). Such variation does not make a noticeable difference in our setting.

## 6 Conclusion

In this work, we reveal the profound connection between layer normalization and the label shift problem in federated learning. We take a first step towards understanding this connection, by identifying the key mechanism of LN as feature normalization. FN simply normalizes the last layer with a scale, but is extremely helpful in addressing feature collapse and local overfitting. Under various heavy label shift settings, the advantage of LN/FN is significant compared to other state-of-the-art algorithms. Some future directions include understanding this connection with more solid theory and testing our finding with a wider range of modalities.

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

## A  Limitations

There are several limitations of our work. First, we did not try to propose a new algorithm, but rather to analyze and understand existing normalization methods. The main contribution is to unveil the connection between normalization and label shift in FL, and to understand this connection.

Moreover, we did not provide a rigorous proof on the convergence rate, since the loss landscape of neural networks is non-convex and it is a longstanding problem in deep learning to rigorously analyze the optimization in usual settings. Rather, we take an alternative approach using feature collapse and local overfitting to qualitatively explain the learning procedures. A more solid foundation for the optimization of normalized networks is an interesting future direction.

On the empirical side, we did not run our experiments several times for the reproducibility check, due to the expensive computation in FL tasks with heavy label shift. On our V100 GPUs, it takes a few days to finish one experiment with 10,000 global rounds and 10 local steps. However, the trend is consistent that normalization methods are much faster in heavy label shift problems, from our experiments on multiple datasets and hyperparameter choices. Last but not least, we have only tested our methods in vision datasets. It would also be interesting to test normalization on other data formats, such as natural language, molecule structure, etc.

## B  Detailed Theorems and Proofs

We first verify the necessity of Assumption 1 here for the scale equivariance of the embedding from the second layer. The necessity of the bias terms should be straightforward. For the activation function, we observe the following lemma:

**Lemma 1** (scale equivariant activation)**.** *If $\rho : \mathbb{R} \to \mathbb{R}$, then $\rho(\lambda t) = \lambda \rho(t)$ for any $\lambda > 0$ and $t \in \mathbb{R}$ iff $\rho$ is a piecewise linear function, with the form:*

$$\rho(t) = \begin{cases} at & t > 0, \\ bt & t \leq 0, \end{cases}, \text{ where } a \text{ and } b \text{ are two real numbers.} \tag{12}$$

Constants $a, b$ can in fact be absorbed into the adjacent layers, allowing Leaky ReLU networks to express all neural networks with activation functions like eq. 12.

*Proof.* Taking $t = 1$, we have $\rho(\lambda) = \lambda \rho(1)$ for $\lambda > 0$. Similarly, $\rho(-\lambda) = \rho(-1)\lambda$. Therefore, $a = \rho(1)$ and $b = \rho(-1)$ are necessary. It remains to prove that $\rho(0) = 0$, which can be obtained with $\lambda = 2$, $t = 0$ in $\rho(\lambda t) = \lambda \rho(t)$. It is also easy to verify that eq. 12 suffices. □

**Proposition 1** (**reduced feature normalization**). *Under Assumption [1], scale normalizing each layer is equivalent to only scale normalizing the last layer. That is, for all affine transformations $\boldsymbol{A}_1, \ldots, \boldsymbol{A}_L$ the function*

$$\mathsf{n} \circ \rho \circ \boldsymbol{A}_L \circ \mathsf{n} \circ \rho \circ \boldsymbol{A}_{L-1} \ldots \mathsf{n} \circ \rho \circ \boldsymbol{A}_1$$

*is equal to*

$$\mathsf{n} \circ \rho \circ \boldsymbol{A}_L \circ \rho \circ \boldsymbol{A}_{L-1} \ldots \rho \circ \boldsymbol{A}_1 \tag{7}$$

*if all the intermediate hidden vectors after activation are non-zero.*

Before we start the proof, we first provide a more formal description of the proposition above.

**Proposition 1'** (**reduced feature normalization**). *Scale normalizing each layer is equivariant to only scale normalizing the last layer. More formally, suppose $\boldsymbol{a}_0 = \boldsymbol{x}$ and $\boldsymbol{g_\theta}(\boldsymbol{x})$ is computed from*

$$\boldsymbol{a}_i = \mathsf{n} \circ \rho(\boldsymbol{U}_i \boldsymbol{a}_{i-1} + \boldsymbol{b}_i), \boldsymbol{U}_i \in \mathbb{R}^{d_i \times d_{i-1}} \text{ for } i \in [L], \boldsymbol{b}_1 \in \mathbb{R}^{d_1}, \boldsymbol{b}_i = \boldsymbol{0}, \text{ for } i \in [2, L]. \tag{13}$$

*If $\rho(\boldsymbol{U}_1 \boldsymbol{a}_0 + \boldsymbol{b}_1) \neq \boldsymbol{0}$ and $\rho(\boldsymbol{U}_i \boldsymbol{a}_{i-1}) \neq \boldsymbol{0}$ for any $i = 2, \ldots, L$, then $\boldsymbol{g_\theta}(\boldsymbol{x})$ is the same as $\boldsymbol{g'_\theta}(\boldsymbol{x})$, with:*

$$\boldsymbol{a}'_L = \boldsymbol{g'_\theta}(\boldsymbol{x}) := \mathsf{n} \circ \rho(\boldsymbol{U}_L \boldsymbol{a}'_{L-1} + \boldsymbol{b}_L), \; \boldsymbol{a}'_i = \rho(\boldsymbol{U}_i \boldsymbol{a}'_{i-1} + \boldsymbol{b}_i), \text{ for } i \in [L-1], \tag{14}$$

*$\boldsymbol{b}_i = \boldsymbol{0}$ for $i = 2, \ldots, L$ and $\boldsymbol{a}'_0 = \boldsymbol{x}$.*

*Proof.* For $L = 1$ it is easy to verify. If $L \geq 2$, we show that there exists $\lambda_1 > 0, \ldots, \lambda_i > 0$ such that $\boldsymbol{a}_i = \lambda_i \boldsymbol{a}'_i$ for $1 \leq i < L$. For $i = 1$, we have $\boldsymbol{a}_1 = \boldsymbol{a}'_1 / \|\boldsymbol{a}'_1\| = \lambda_1 \boldsymbol{a}'_1$ with $\lambda_1 = 1/\|\boldsymbol{a}'_1\|$. Suppose for $1 \leq i < L-1$, $\boldsymbol{a}_i = \lambda_i \boldsymbol{a}'_i$ holds. Then

$$\boldsymbol{a}_{i+1} = \mathsf{n} \circ \rho(\boldsymbol{U}_{i+1} \boldsymbol{a}_i) = \frac{\rho(\boldsymbol{U}_{i+1} \boldsymbol{a}_i)}{\|\rho(\boldsymbol{U}_{i+1} \boldsymbol{a}_i)\|} = \frac{\rho(\boldsymbol{U}_{i+1} \lambda_i \boldsymbol{a}'_i)}{\|\rho(\boldsymbol{U}_{i+1} \lambda_i \boldsymbol{a}'_i)\|} = \frac{\rho(\boldsymbol{U}_{i+1} \boldsymbol{a}'_i)}{\|\rho(\boldsymbol{U}_{i+1} \boldsymbol{a}'_i)\|} = \lambda_{i+1} \boldsymbol{a}'_{i+1}, \tag{15}$$

where $\lambda_{i+1} = 1/\|\boldsymbol{a}'_{i+1}\|$. Therefore, from $\boldsymbol{a}_{L-1} = \lambda_{L-1} \boldsymbol{a}'_{L-1}$ we obtain:

$$\boldsymbol{a}_L = \mathsf{n} \circ \rho(\boldsymbol{U}_L \boldsymbol{a}_{L-1}) = \frac{\rho(\boldsymbol{U}_L \boldsymbol{a}_{L-1})}{\|\rho(\boldsymbol{U}_L \boldsymbol{a}_{L-1})\|} = \frac{\rho(\boldsymbol{U}_L \boldsymbol{a}'_{L-1})}{\|\rho(\boldsymbol{U}_L \boldsymbol{a}'_{L-1})\|} = \mathsf{n} \circ \rho(\boldsymbol{U}_L \boldsymbol{a}'_{L-1}) = \boldsymbol{a}'_L. \tag{16}$$

$\square$

**Proposition 2** (**reduced layer normalization**). *Under Assumption [1], MV normalizing each layer is equivalent to only MV normalizing the last layer and shifting previous layers. That is, for all affine transformations $\boldsymbol{A}_1, \ldots, \boldsymbol{A}_L$ the function*

$$\mathsf{n_{MV}} \circ \rho \circ \boldsymbol{A}_L \circ \mathsf{n_{MV}} \circ \rho \circ \boldsymbol{A}_{L-1} \ldots \mathsf{n_{MV}} \circ \rho \circ \boldsymbol{A}_1$$

*is equal to*

$$\mathsf{n_{MV}} \circ \rho \circ \boldsymbol{A}_L \circ \mathsf{s} \circ \rho \circ \boldsymbol{A}_{L-1} \cdots \circ \mathsf{s} \circ \rho \circ \boldsymbol{A}_1 \tag{8}$$

*if none of the intermediate hidden layer activations to normalize are proportional to the all-one vector $\boldsymbol{1}$.*

We similarly provide a more formal description.

**Proposition 2'** (**reduced layer normalization**). *Suppose our layer-normalized neural network is:*

$$\boldsymbol{a}_i = \mathsf{n_{MV}} \circ \rho(\boldsymbol{U}_i \boldsymbol{a}_{i-1} + \boldsymbol{b}_i), \boldsymbol{U}_i \in \mathbb{R}^{d_i \times d_{i-1}} \text{ for } i \in [L], \boldsymbol{b}_1 \in \mathbb{R}^{d_1}, \boldsymbol{b}_i = \boldsymbol{0}, \text{ for } i \in [2, L]. \tag{17}$$

*with $\boldsymbol{a}_0 = \boldsymbol{x}, \boldsymbol{a}_L = \boldsymbol{g_\theta}(\boldsymbol{x})$, and $\rho(\boldsymbol{U}_i \boldsymbol{a}_{i-1} + \boldsymbol{b}_i) \neq \mu(\rho(\boldsymbol{U}_i \boldsymbol{a}_{i-1} + \boldsymbol{b}_i))\boldsymbol{1}$ for $i \in [L]$. Then $\boldsymbol{g_\theta}(\boldsymbol{x})$ as computed from eq. [17], is the same as $\boldsymbol{g'_\theta}(\boldsymbol{x})$, computed from:*

$$\boldsymbol{a}'_L = \boldsymbol{g'_\theta}(\boldsymbol{x}) := \mathsf{n_{MV}} \circ \rho(\boldsymbol{U}_L \boldsymbol{a}'_{L-1} + \boldsymbol{b}_L), \; \boldsymbol{a}'_i = \mathsf{s} \circ \rho(\boldsymbol{U}_i \boldsymbol{a}'_{i-1} + \boldsymbol{b}_i), \text{ for } i \in [L-1], \tag{18}$$

*$\boldsymbol{b}_i = \boldsymbol{0}$ for $i = 2, \ldots, L$ and $\boldsymbol{a}'_0 = \boldsymbol{x}$.*

*Proof.* The proof follows analogously from the proof of Proposition [1']. $\square$

### B.1 Expressive power of normalized networks

In this subsection, we provide the proof steps for Proposition 3. First, we recall:

$$
\begin{aligned}
\varepsilon(\boldsymbol{\theta}, \boldsymbol{W}; \boldsymbol{x}, y) &= \chi(y \notin \mathrm{argmax}_i \, \hat{y}_i), \quad \text{with } \hat{\boldsymbol{y}} = \mathrm{softmax}(\boldsymbol{W} \boldsymbol{g}_{\boldsymbol{\theta}}(\boldsymbol{x})), \\
\varepsilon_F(\boldsymbol{\theta}, \boldsymbol{W}; \boldsymbol{x}, y) &= \chi(y \notin \mathrm{argmax}_i \, \hat{y}_i), \quad \text{with } \hat{\boldsymbol{y}} = \mathrm{softmax}(\boldsymbol{W} \, \mathsf{n} \circ \boldsymbol{g}_{\boldsymbol{\theta}}(\boldsymbol{x})),
\end{aligned}
\tag{19}
$$

where $\chi$ is the indicator function. We prove the first part of Proposition 3 for feature normalized networks:

**Proposition 3.A** (expressive power of FN)**.** *For any model parameters $(\boldsymbol{\theta}, \boldsymbol{W})$ and any sample $(\boldsymbol{x}, y)$ with $\boldsymbol{g}_{\boldsymbol{\theta}}(\boldsymbol{x}) \neq \boldsymbol{0}$, we have $\varepsilon(\boldsymbol{\theta}, \boldsymbol{W}; \boldsymbol{x}, y) = \varepsilon_F(\boldsymbol{\theta}, \boldsymbol{W}; \boldsymbol{x}, y)$.*

*Proof.* We first show:

$$
\varepsilon(\boldsymbol{\theta}, W; \boldsymbol{x}, y) = 0 \implies \varepsilon_F(\boldsymbol{\theta}, W; \boldsymbol{x}, y) = 0.
\tag{20}
$$

If the l.h.s. is true, then $\boldsymbol{w}_y^\top \boldsymbol{g}_{\boldsymbol{\theta}} \geq \boldsymbol{w}_c^\top \boldsymbol{g}_{\boldsymbol{\theta}}$ for any $c$. This gives

$$
\frac{\boldsymbol{w}_y^\top \boldsymbol{g}_{\boldsymbol{\theta}}}{\|\boldsymbol{g}_{\boldsymbol{\theta}}\|} \geq \frac{\boldsymbol{w}_c^\top \boldsymbol{g}_{\boldsymbol{\theta}}}{\|\boldsymbol{g}_{\boldsymbol{\theta}}\|}, \quad \text{for any } c.
$$

From eq. 19 we know $\varepsilon_F(\boldsymbol{\theta}, W; \boldsymbol{x}, y) = 0$. Similarly, we can show

$$
\varepsilon_F(\boldsymbol{\theta}, W; \boldsymbol{x}, y) = 0 \implies \varepsilon(\boldsymbol{\theta}, W; \boldsymbol{x}, y) = 0.
\tag{21}
$$

This concludes the proof of FN. $\qquad\square$

Note that the assumption $\boldsymbol{g}_{\boldsymbol{\theta}} \neq \boldsymbol{0}$ can be removed if we use $\max\{\epsilon, \|\boldsymbol{g}_{\boldsymbol{\theta}}\|\}$ in our feature normalization. Similarly, we can show that pre-activation/post-activation layer normalization is not more expressive than vanilla models. From Prop. 2, a pre-activation LN network can be represented as:

$$
\boldsymbol{a}_L = \boldsymbol{g}_{\boldsymbol{\theta}}(\boldsymbol{x}) := \rho \circ \mathsf{n}_{\mathsf{MV}}(\boldsymbol{U}_L \boldsymbol{a}_{L-1} + \boldsymbol{b}_L), \; \boldsymbol{a}_i = \rho \circ \mathsf{s}(\boldsymbol{U}_i \boldsymbol{a}_{i-1} + \boldsymbol{b}_i), \; \text{for } i \in [L-1],
\tag{22}
$$

with $\boldsymbol{b}_i = \boldsymbol{0}$ for $i = 2, \ldots, L$ and $\boldsymbol{a}_0 = \boldsymbol{x}$. We use $\varepsilon_{pL}$ to denote the accuracy function of pre-activation LN, and $\varepsilon_{Lp}$ to denote post-activation LN.

**Proposition 3.B** (expressive power of LN)**.** *For any model parameter $(\boldsymbol{\theta}, \boldsymbol{W})$ of pre-activation layer normalization, one can find $(\boldsymbol{\theta}', \boldsymbol{W})$ of a vanilla model such that $\varepsilon(\boldsymbol{\theta}', \boldsymbol{W}; \boldsymbol{x}, y) = \varepsilon_{pL}(\boldsymbol{\theta}, \boldsymbol{W}; \boldsymbol{x}, y)$, for any $\boldsymbol{x}, y$. For any model parameter $(\boldsymbol{\theta}, \boldsymbol{W})$ of post-activation layer normalization, one can find $(\boldsymbol{\theta}', \boldsymbol{W}')$ of a vanilla model such that $\varepsilon(\boldsymbol{\theta}', \boldsymbol{W}'; \boldsymbol{x}, y) = \varepsilon_{Lp}(\boldsymbol{\theta}, \boldsymbol{W}; \boldsymbol{x}, y)$, for any $\boldsymbol{x}, y$.*

*Proof.* For the proof of LN networks, first observe that pre-activation layer normalization is equivalent to eq. 22. Each layer of the form $(i = 1, 2, \ldots, L)$:

$$
\boldsymbol{a}_i = \rho \circ \mathsf{s}(\boldsymbol{U}_i \boldsymbol{a}_{i-1} + \boldsymbol{b}_i),
\tag{23}
$$

is equivalent to:

$$
\boldsymbol{a}_i = \rho \left( \boldsymbol{P}_i \boldsymbol{U}_i \boldsymbol{a}_{i-1} + \boldsymbol{P}_i \boldsymbol{b}_i \right),
\tag{24}
$$

where $\boldsymbol{P}_i = \boldsymbol{I} - \frac{1}{d_i} \boldsymbol{1} \boldsymbol{1}^\top$ is a projection matrix and $d_i$ is the dimension of $\boldsymbol{a}_i$. Thus, we can take $\boldsymbol{U}_i' = \boldsymbol{P}_i \boldsymbol{U}_i$ and $\boldsymbol{b}_i' = \boldsymbol{P}_i \boldsymbol{b}_i$ so that:

$$
\rho \circ \mathsf{s}(\boldsymbol{U}_i \boldsymbol{a}_{i-1} + \boldsymbol{b}_i) = \rho(\boldsymbol{U}_i' \boldsymbol{a}_{i-1} + \boldsymbol{b}_i').
\tag{25}
$$

Also note that the normalization factor in the last layer does not affect the final prediction, as seen from Proposition 3.A. Therefore, collecting all the $\boldsymbol{U}_i'$ and $\boldsymbol{b}_i'$ for $i = 1, 2, \ldots, L$ we obtain the required $(\boldsymbol{\theta}', \boldsymbol{W})$.

We can show a similar conclusion for post-activation LN. Suppose $(\boldsymbol{U}_1, \boldsymbol{U}_2, \ldots, \boldsymbol{U}_L, \boldsymbol{b}_1, \boldsymbol{W})$ are the parameters of an LN network. We can first take $\boldsymbol{U}_1' = \boldsymbol{U}_1$, $\boldsymbol{b}_1' = \boldsymbol{b}_1$ such that:

$$\boldsymbol{z}_1 := \rho(\boldsymbol{U}_1 \boldsymbol{x} + \boldsymbol{b}_1) = \boldsymbol{a}_1, \tag{26}$$

where $\boldsymbol{a}_1$ is the output of the first layer of a vanilla network. Denote $\boldsymbol{a}_i' = \mathsf{s}(\boldsymbol{z}_i)$ and $\boldsymbol{z}_i = \rho(\boldsymbol{U}_i \boldsymbol{a}_{i-1}' + \boldsymbol{b}_i)$ for $i \geq 1$. Assume $\boldsymbol{z}_{i-1} = \boldsymbol{a}_{i-1}$ where $\boldsymbol{a}_i$ is the output of the $i^{\text{th}}$ layer of a vanilla network with parameters $(\boldsymbol{U}_1', \boldsymbol{U}_2', \ldots, \boldsymbol{U}_L', \boldsymbol{b}_1', \boldsymbol{W}')$, then we can prove that for $i \geq 2$:

$$\boldsymbol{z}_i = \rho(\boldsymbol{U}_i \boldsymbol{a}_{i-1}') = \rho(\boldsymbol{U}_i \mathsf{s}(\boldsymbol{z}_{i-1})) = \rho(\boldsymbol{U}_i \boldsymbol{P}_{i-1} \boldsymbol{a}_{i-1}) = \rho(\boldsymbol{U}_i' \boldsymbol{a}_{i-1}) = \boldsymbol{a}_i, \tag{27}$$

if we take $\boldsymbol{U}_i' = \boldsymbol{U}_i \boldsymbol{P}_{i-1}$. Therefore, by induction, we can prove that $\boldsymbol{z}_i = \boldsymbol{a}_i$ for $i = 1, 2, \ldots, L$. Specifically, we have $\boldsymbol{z}_L = \rho(\boldsymbol{U}_L \boldsymbol{a}_{L-1}' + \boldsymbol{b}_L) = \boldsymbol{a}_L$.

Since $\varepsilon_{Lp}(\boldsymbol{\theta}, \boldsymbol{W}; \boldsymbol{x}, y) = 0$ iff:

$$\boldsymbol{w}_y^\top \boldsymbol{a}_L' \geq \boldsymbol{w}_c^\top \boldsymbol{a}_L', \text{ for all } c \in [C], \tag{28}$$

and $\boldsymbol{a}_L' = \mathsf{n}_{\mathsf{MV}}(\boldsymbol{z}_L) = \frac{\mathsf{s}(\boldsymbol{z}_L)}{\sigma(\boldsymbol{z}_L)} = \boldsymbol{P}_L \boldsymbol{z}_L / \sigma(\boldsymbol{z}_L)$, with $\boldsymbol{P}_L = \boldsymbol{I} - \frac{1}{d_L} \mathbf{1}\mathbf{1}^\top$, eq. 28 is equivalent to:

$$\boldsymbol{w}_y^\top \boldsymbol{P}_L \boldsymbol{a}_L \geq \boldsymbol{w}_c^\top \boldsymbol{P}_L \boldsymbol{a}_L, \text{ for all } c \in [C]. \tag{29}$$

Hence, taking $\boldsymbol{W}' = \boldsymbol{P}_L \boldsymbol{W}$ gives the desired vanilla network. $\qquad\square$

Next, we can prove the following corollary:

**Corollary 1.** *Suppose $\varepsilon^*$, $\varepsilon_F^*$ and $\varepsilon_L^*$ are the optimal prediction errors, by vanilla, feature-normalized and (pre/post-activation) layer-normalized neural networks respectively on a fixed dataset $S$:*

$$\varepsilon_{\varnothing, F, L}^* = \inf_{\boldsymbol{\theta}, \boldsymbol{W}} \mathbb{E}_{(\boldsymbol{x}, y) \in S}\, \varepsilon_{\varnothing, F, L}(\boldsymbol{\theta}, \boldsymbol{W}; \boldsymbol{x}, y), \tag{30}$$

*Then we have: $\varepsilon^* = \varepsilon_F^* \leq \varepsilon_L^*$. Note that $\varepsilon_\varnothing$ has the same meaning as $\varepsilon$.*

*Proof.* Suppose for an LN network, $(\boldsymbol{\theta}_i^*, \boldsymbol{W}_i^*)$ is a sequence that can achieve prediction error $\varepsilon_L^* + \delta_i$ with $\delta_i \downarrow 0$. Then from Proposition 3.B, there exists $(\boldsymbol{\theta}_i'^*, \boldsymbol{W}_i'^*)$ of a vanilla network which can achieve prediction error $\varepsilon_L^* + \delta_i$ and $\delta_i \downarrow 0$, and thus $\varepsilon^* \leq \varepsilon_L^*$. Similarly, we can show $\varepsilon^* = \varepsilon_F^*$ from Proposition 3.A. $\qquad\square$

## B.2 Proof of divergent norms

**Theorem 1** (**divergent norms**). *In order to minimize the one-class local loss $f_k(\boldsymbol{\theta}, \boldsymbol{W})$, we must have at least one of the following: 1) $\|\boldsymbol{w}_c\| \to \infty$ for all $c \in [C]$ and $c \neq k$; 2) $\|\boldsymbol{g}_{\boldsymbol{\theta}}(\boldsymbol{x}_i)\| \to \infty$ for all $\boldsymbol{x}_i$; 3) $\|\boldsymbol{w}_k\| \to \infty$.*

*Proof.* We first show that it is possible and necessary to have $(\boldsymbol{w}_c - \boldsymbol{w}_k)^\top \boldsymbol{g}_{\boldsymbol{\theta}}(\boldsymbol{x}_i) < 0$ for all $\boldsymbol{x}_i \in S_k$ and $c \neq k$. Suppose the $j^{\text{th}}$-layer affine transformation is $\boldsymbol{A}_j(\boldsymbol{a}) = \boldsymbol{U}_j \boldsymbol{a} + \boldsymbol{b}_j$, with $\boldsymbol{U}_j \in \mathbb{R}^{d_j \times d_{j-1}}$ and $\boldsymbol{b}_j \in \mathbb{R}^{d_j}$.

Since $\rho \neq 0$, there exists $x_0 \in \mathbb{R}$ such that $\rho(x_0) \neq 0$. For the possibility, by taking $\boldsymbol{U}_1 = \boldsymbol{0}$, $\boldsymbol{b}_1 = x_0 \mathbf{1}$ and $\boldsymbol{U}_j = \frac{x_0}{\rho(x_0) d_{j-1}} \mathbf{1}\mathbf{1}^\top$, $\boldsymbol{b}_j = \boldsymbol{0}$ for $j \geq 2$, we have $\boldsymbol{a}_j = \rho(x_0)\mathbf{1}$ for all $j \geq 1$, and specifically $\boldsymbol{g}_{\boldsymbol{\theta}} = \boldsymbol{a}_L = \rho(x_0)\mathbf{1}$. If we choose $\boldsymbol{w}_k = \rho(x_0)\mathbf{1}$ and $\boldsymbol{w}_c = -\rho(x_0)\mathbf{1}$ for all $c \neq k$, then $(\boldsymbol{w}_c - \boldsymbol{w}_k)^\top \boldsymbol{g}_{\boldsymbol{\theta}}(\boldsymbol{x}_i) < 0$ can be achieved. Moreover, by replacing $\boldsymbol{W}$ with $t\boldsymbol{W}$ and taking $t \to \infty$, the loss $f_k(\boldsymbol{\theta}, \boldsymbol{W})$ can be arbitrarily close to zero.

For the necessity, suppose otherwise $(\boldsymbol{w}_c - \boldsymbol{w}_k)^\top \boldsymbol{g}_{\boldsymbol{\theta}}(\boldsymbol{x}_i) \geq 0$ for some $\boldsymbol{x}_i \in S_k$ and $c \neq k$. Then $f_k(\boldsymbol{\theta}, \boldsymbol{W}) \geq \frac{1}{m_k} \log C$. However, we have seen in the last paragraph that the infimum of $f_k$ is zero.

Now suppose $(\boldsymbol{w}_c - \boldsymbol{w}_k)^\top \boldsymbol{g}_{\boldsymbol{\theta}}(\boldsymbol{x}_i) < 0$ for all $\boldsymbol{x}_i \in S_k$ and $c \neq k$ and all the three claims are false. Then $\|\boldsymbol{w}_k\|$ is bounded, $\|\boldsymbol{w}_c\|$ is bounded for some $c \neq k$, and $\|\boldsymbol{g}_{\boldsymbol{\theta}}(\boldsymbol{x}_i)\|$ is bounded for some $\boldsymbol{x}_i$. So $f_k(\boldsymbol{\theta}, \boldsymbol{W}) \geq \ell(\boldsymbol{\theta}, \boldsymbol{W}; \boldsymbol{x}_i, k)/m_k$ is lower bounded. This is a contradiction. $\qquad\square$

## C   Experimental Setup

In this appendix, we provide details for our experimental setup.

### C.1   Datasets and models

To evaluate the performance of our proposed method we use common FL benchmarks. We use four datasets: CIFAR-10/100 (Krizhevsky et al., 2009), TinyImageNet (Le & Yang, 2015), and PACS (Li et al., 2017). For CIFAR-100, we use the CNN network (Fukushima, 1975), and for CIFAR-10, we test both CNN and ResNet-18 (He et al., 2016). For the remaining datasets, we use ResNet-18. Since we are investigating the effect of normalization on the performance of the FL, we use ResNet-18 without any normalization (i.e., CNN + skip connection) as our base method. In ResNet-18, we add layer norm (LN), group norm (GN, Wu & He, 2018) or batch norm (BN) after each activation, and in ResNet-18 with feature norm (FN), we only apply scale normalization to the output features of the network before the last fully connected layer. Similarly, for CNN models, we add normalization methods after each activation. For all the normalization methods including FN, we set $\epsilon = 1e^{-5}$ for stability (see the preliminaries). For the implementation of LN/GN/BN, we may track the running statistics and average all the parameters including the running means and running variances.

Table 7: CNN model for CIFAR-10 dataset. we denote $k$ as the kernel size, $c$ as the number of channels, and $s$ as the size of the stride.

| Layer | Models | | | |
|---|---|---|---|---|
| | Vanilla | LN | BN | FN |
| Input (shape) | (28, 28, 3) | (28, 28, 3) | (28, 28, 3) | (28, 28, 3) |
| Conv2d $(k, c, s)$ | (5, 64, 1) | (5, 64, 1) | (5, 64, 1) | (5, 64, 1) |
| Normalization | - | layer norm | batch norm | - |
| MaxPool2d | (2, 2) | (2, 2) | (2, 2) | (2, 2) |
| Conv2d $(k, c, s)$ | (5, 64, 1) | (5, 64, 1) | (5, 64, 1) | (5, 64, 1) |
| Normalization | - | layer norm | batch norm | - |
| MaxPool2d | (2, 2) | (2, 2) | (2, 2) | (2, 2) |
| Flatten (shape) | 1600 | 1600 | 1600 | 1600 |
| Dense (in $c$, out $c$) | (1600, 384) | (1600, 384) | (1600, 384) | (1600, 384) |
| Normalization | - | layer norm | batch norm | feature norm |
| Dense (in $c$, out $c$) | (384, 10/100) | (384, 10/100) | (384, 10/100) | (384, 10/100) |

In order to preserve the scale invariance (equivariance), we remove all the biases for the models. We summarize the CNN model structure with different normalization layers in Table 7. We use the same architecture except for the last layer for both CIFAR-10 and CIFAR-100. We keep ReLU as our activation function, and softmax for the last layer.

### C.2   Data partitioning

Our client datasets are created through partitioning common datasets including CIFAR-10, CIFAR-100, TinyImageNet and PACS. We consider two types of data partitioning to simulate label shift in practice:

- $n$ class(es): in this setting, each client has access to samples from only $n$ classes.
- Dirichlet partitioning: we partition samples from each class with a symmetric Dirichlet distribution and parameter $\beta$. For each client, we collect one partitioned shard from each class. We denote this with the shorthand notation Dir($\beta$). See Zhang et al. (2023) as an example.

Our setup includes 10 clients for the CIFAR-10 dataset, 50 and 20 clients for the CIFAR-100 dataset, 200 clients for TinyImageNet, and 12 clients for PACS. We distribute the entire training and test datasets among clients in a non-i.i.d. manner. More specifically, we use one class per client, two classes per client,

Table 8: Dataset distribution for datasets.

| Dataset | Train clients | Train examples | Test clients | Test examples | # of classes |
|---|---|---|---|---|---|
| CIFAR-10 | 10 | 50,000 | 10 | 10,000 | 10 |
| CIFAR-100 | 50/20 | 50,000 | 50/20 | 10,000 | 100 |
| TinyImageNet | 20 | 25,000 | 20 | 25,000 | 200 |
| PACS | 12 | 7,907 | 12 | 1,920 | 7 |

Figure 8: Data distribution of CIFAR-10 dataset across clients with label shift. (**left**) one-class: each client has access to one-class; (**middle**) two classes: each client has access to two classes; (**right**) Dirichlet (0.1): Dirichlet allocation with $\beta = 0.1$.

and Dirichlet allocation with $\beta = 0.1$ for the CIFAR-10 dataset as shown in Figure 8. For the CIFAR-100 dataset, we use two classes per client (50 clients), 5 classes per client (20 clients), and Dirichlet allocation with $\beta = 0.1$ (20 clients). For TinyImageNet, we apply Dirichlet allocation with $\beta = 0.1, 0.2, 0.5$.

Finally, for the PACS dataset, to add distribution shift to the clients, we first split the dataset into 4 groups, photo (P), art painting (A), clipart (C) and sketch (S). For each group, we split the data into 3 clients, with Dirichlet partitioning Dir(0.5) and Dir(1.0), as well as partitioning into disjoint sets of classes (two clients have two classes each, and the other client has three classes of samples).

Statistics on the number of clients and examples in both the training and test splits of the datasets are given in Table 8. To illustrate the data partitioning, we include the label distributions for each client for CIFAR-10 and PACS. The label distributions of the CIFAR-10 dataset are shown in Figure 8. For the PACS dataset, the label distributions are shown in Figure 9.

## C.3 Optimizers, hyperparameters, and validation metrics

We use SGD with a 0.01 learning rate (`lr`) and batch size of 32 for all of the experiments except for $E = 1$ experiments in CIFAR-100 in which we take `lr = 0.1` as the learning rate and `lr = 0.001` for PACS. We use SGD with a momentum of 0.9 only for our centralized training baseline. In each client, we take $E$ steps of local training in which we iterate $E$ batches of data per client. We use online augmentation with random horizontal flip and random cropping with padding of 4 pixels for all of the datasets. Moreover, we test and utilize FedYogi (Reddi et al., 2020) as a server adaptive optimizer method in combination with FN. All the reported experiments are done with 10,000 global rounds. We report the accuracy of clients on their test data in the last iteration of our experiments with a single run due to computational limitation. Our hyperparameter choices are summarized in Table 9.

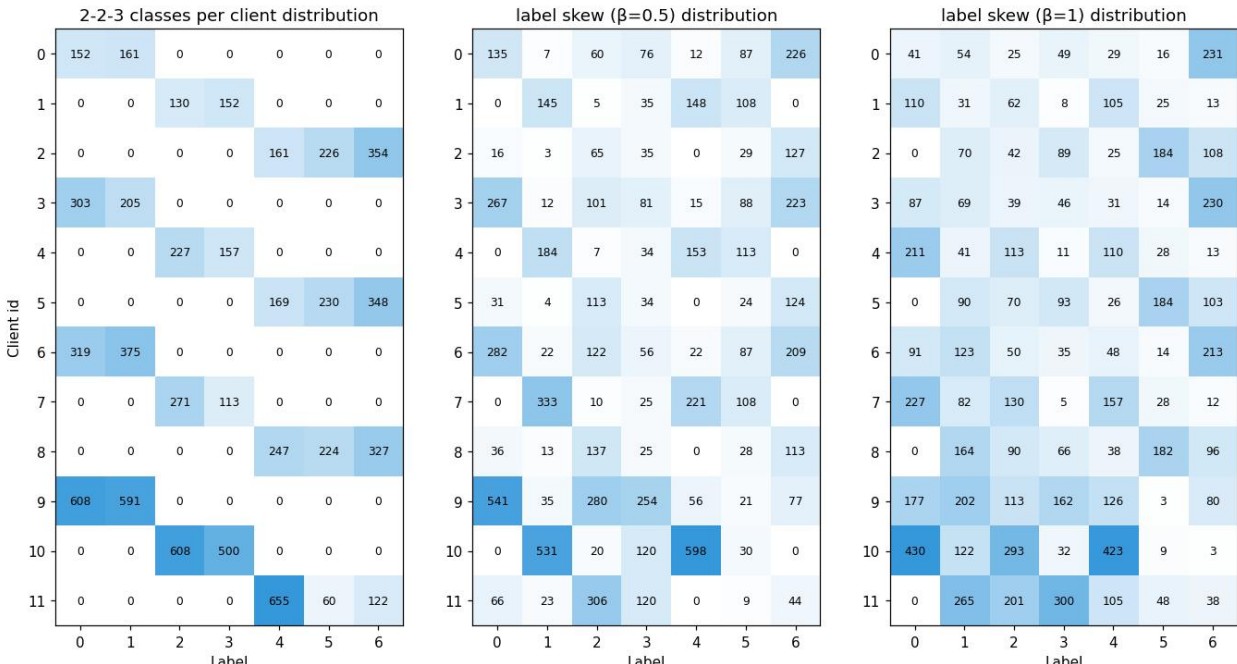

Figure 9: Data distribution of the PACS dataset across clients with label shift. (**left**) 2-2-3 classes: each client has access to two classes except the last client who has access to three classes; (**middle**) Dirichlet allocation with $\beta = 0.5$; (**right**) Dirichlet allocation with $\beta = 1$.

Table 9: Hyper-parameter choices.

| Dataset | CIFAR-10 (CNN) | CIFAR-10 (ResNet) | CIFAR-100 | PACS | TinyImageNet |
|---|---|---|---|---|---|
| Learning rate | 0.01 | 0.01 | 0.1 | 0.001 | 0.01 |
| Batch size | 32 | 32 | 32 | 32 | 32 |
| Optimizer | SGD | SGD | SGD | SGD | SGD |
| Augmentation | ✓ | ✓ | ✓ | ✓ | ✓ |

## C.4 Benchmarks

In this section, we provide the implementation details and hyperparameters of our benchmark experiments.

**FedProx.** For the implementation of FedProx, as stated in Li et al. (2020a), we add a weighted regularizer to the loss function of each client during local training to penalize the divergence between the server model and clients' models. To set the weight of the regularizer (or $\mu$ in the original paper), we sweep the datasets for weights in $0.001, 0.01, 0.1, 1$ and select the best-performing value in the Dirichlet distribution with $E = 20$ for each dataset. Our results show that 0.01 was the best-performing value for the weights for all the datasets. For the rest of the hyperparameters, we use the hyperparameters as FedAvg.

**SCAFFOLD.** For the implementation of SCAFFOLD, we use Option II as described in Karimireddy et al. (2020). We use the same setting and hyperparameters as FedAvg. More specifically, clients perform $E$ steps of local training while clients control variates using the difference between the server model and their local learned model.

**FedDecorr.** Similar to FedProx, FedDecorr (Shi et al., 2023) applies a weighted regularization term during local training that encourages different dimensions of representations to be uncorrelated. We utilize their code for the implementation. In their results, the best-performing weight for the regularization term (or $\alpha$

in the original paper) was 0.5. Therefore, we set the weight to 0.5 for all the datasets. For the rest of the hyperparameters, we use the same setting and hyperparameters as FedAvg.

**FedLC.** As described in Zhang et al. (2022), for the implementation of FedLC, during the local training, we shift the output logits of the network for each class depending on the number of available samples of each class for that client. This allows clients to add a weighted pairwise label margin according to the probability of occurrence of each class. We set the weight (or $\tau$ in the original paper) to 1. As shown in Table 1, FedLC fails in the one-class setting since clients do not have two classes to create a label margin. For the rest of the hyperparameters, we use the same setting and hyperparameters as FedAvg.

**FedRS.** Similar to FedLC, FedRS (Li & Zhan, 2021) attempts to address the label shift in clients by limiting the update of missing classes' weights during the local training. FedRS multiplies the output logits of the missing classes by a weighting parameter ($\alpha$ in the original paper) and keeps the rest of the logits intact. As described in Li & Zhan (2021), 0.5 was the best-performing weight for all the datasets, and therefore, we set the weight to 0.5 for our experiments. For the rest of the hyperparameters, including the learning rate, we use the same hyperparameters as FedAvg.

**FedYogi** is one of the server-side adaptive optimization methods described in Reddi et al. (2020). Basically, it utilizes a separate learning rate for the server to calculate the global model for the clients. We utilize the best-performing values for the CIFAR-10 experiments in Reddi et al. (2020). More specifically, we set the server learning rate to 0.01 ($\eta$ in the original paper), $\beta_1$, $\beta_2$, and $\tau$ to 0.9, 0.99, and 0.001 respectively. We also initialize vector $v_t$ to $1e^{-6}$. For the rest of the hyperparameters, including the clients' learning rates, we use the same hyperparameters as FedAvg.

### C.5 Hardware

Our experiments are run on a cluster of 12 GPUs, including NVIDIA V100 and P100.

## D Experiment Configurations for Figures

We provide details for the experimental setup of the figures in our main paper. Unless otherwise specified, we fix the local step number $E = 10$ in our experiments.

**Figure 2.** In this experiment, we first train a vanilla CNN model with CIFAR-10 dataset in the one-class setting using FedAvg for 10000 epochs with 10 local steps and report the class accuracy of the global model (**left**) using the same setup as rest of the paper. Then, we locally train this model using examples belonging to class 0 (i.e. data of client 0) for 5 steps with FN (**right**) and without FN (**middle**) and report the class accuracy of these two models in the entire dataset. Note that FN networks and vanilla networks are the same except the constraint on the feature embedding.

**Figure 3.** In this one-class setup, for the three figures on the left, we only locally train client 1 with its local data which only contains class 1.

**Figure 4.** We use SGD with batch size 32 for FedAvg with 10 local steps and batch size of 320 for centralized training. We use 0.01 as the learning rate for both centralized and FedAvg. The batch size 320 is chosen to match federated and centralized learning.

**Figure 6.** We use batch size 32, learning rate 0.001, and SGD with 0.9 momentum for the centralized training.

## E Additional Experiments

**Additional local steps and different neural architectures.** We provide additional experiments with different local steps, on CIFAR-10/100 and CNN architecture in Tables 10 and 11. This table shows our results are consistent with different local steps. As label shift becomes more severe, the advantage of FN/LN

Table 10: SoTA comparison on CIFAR-10 with CNN.

| Methods | 1 class | | | 2 classes | | | Dir(0.1) | | |
|---|---|---|---|---|---|---|---|---|---|
| | $E = 1$ | $E = 10$ | $E = 20$ | $E = 1$ | $E = 10$ | $E = 20$ | $E = 1$ | $E = 10$ | $E = 20$ |
| FedAvg | 57.45 | 55.86 | 58.65 | 57.35 | 71.83 | 73.16 | 52.82 | 73.06 | 74.58 |
| FedProx | 57.39 | 54.24 | 58.31 | 57.42 | 71.86 | 72.78 | 53.94 | 72.85 | 74.72 |
| SCAFFOLD | 57.41 | 54.14 | 57.97 | 57.45 | 71.88 | 73.44 | 52.87 | 72.38 | 75.20 |
| FedLC | 9.72 | 9.72 | 9.72 | 39.35 | 62.01 | 64.49 | 45.08 | 70.25 | 71.98 |
| FedDecorr | 47.50 | 48.18 | 48.09 | 47.73 | 70.88 | 73.43 | 45.46 | 74.58 | 75.31 |
| FedRS | 10.00 | 10.00 | 10.00 | 37.96 | 63.26 | 64.06 | 44.82 | 70.60 | 72.63 |
| FedYogi | 77.61 | 73.13 | 74.21 | 76.77 | **78.97** | **77.12** | 76.46 | **79.35** | **77.99** |
| FedFN | 77.52 | 77.14 | 77.35 | 74.52 | 76.67 | 76.12 | 75.73 | 78.13 | 77.35 |
| FedLN | **79.25** | **78.04** | **78.12** | **77.39** | 77.52 | 76.93 | **77.10** | 78.27 | 77.48 |

Table 11: SoTA comparison on CIFAR-100 with CNN.

| Methods | 2 classes | | | 5 classes | | | Dir(0.01) | | |
|---|---|---|---|---|---|---|---|---|---|
| | $E = 1$ | $E = 10$ | $E = 20$ | $E = 1$ | $E = 10$ | $E = 20$ | $E = 1$ | $E = 10$ | $E = 20$ |
| FedAvg | 39.38 | 32.33 | 33.57 | 41.81 | 36.22 | 36.65 | 40.92 | 38.14 | 38.12 |
| FedProx | 42.47 | 32.39 | 31.70 | 41.65 | 36.59 | 36.95 | 44.35 | 38.85 | 38.67 |
| SCAFFOLD | 41.40 | 32.30 | 34.43 | 41.46 | 36.57 | 37.06 | 44.55 | 38.7 | 38.69 |
| FedLC | 4.41 | 6.04 | 5.75 | 17.63 | 18.17 | 18.78 | 21.31 | 22.34 | 22.35 |
| FedDecorr | 33.03 | 29.21 | 32.43 | 33.27 | 32.94 | 35.26 | 33.58 | 34.24 | 36.48 |
| FedRS | 6.35 | 10.73 | 10.81 | 22.30 | 24.88 | 25.29 | 21.78 | 22.19 | 24.13 |
| FedYogi | 46.66 | 44.60 | 43.98 | 46.79 | 44.08 | 43.89 | 48.10 | 45.20 | 44.96 |
| FedFN | 46.51 | 40.32 | 43.70 | 45.51 | 45.41 | 44.10 | 48.15 | 45.36 | 44.52 |
| FedLN | **48.72** | **46.59** | **45.89** | **49.83** | **46.22** | **45.68** | **51.21** | **47.30** | **45.53** |

Table 12: Combining adaptive optimization with normalization.

| Methods | 1 class | 2 classes | Dir(0.1) |
|---|---|---|---|
| FedAvg | 56.82 | 71.83 | 73.06 |
| FedYogi | 73.13 | 78.97 | 79.35 |
| FedFN | 77.14 | 76.67 | 78.14 |
| FedLN | 77.71 | 77.65 | 79.94 |
| FedFN + Yogi | **80.47** | 78.83 | 80.35 |
| FedLN + Yogi | 79.27 | **79.16** | **80.46** |

is clearer. Note that $E = 1$ it reduces to the centralized setting with 320 batch size (since each client has batch size 32), but due to label shift not every algorithm can converge well.

**Combination with other techniques.** We can further improve the performance of normalization, by combining it with other techniques such as adaptive optimization (Reddi et al., 2020). In Table 12, we show that combined with FedYogi (Reddi et al., 2020), FedFN and FedLN has improved performance. We do not consider other combinations such as with FedProx or SCAFFOLD as they are not shown to help FedAvg much even in the unnormalized case.

Table 13: Comparison between different modifications of ResNet.

| Methods | 1 class | 2 classes | Dir(0.1) |
|---|---|---|---|
| FedLN - ResNet | 86.15 | 88.01 | 89.06 |
| FedFN - $n_{MV}$ inside - ResNet | 77.76 | 84.34 | 86.63 |
| FedFN - n inside - ResNet | 82.14 | 84.05 | 85.17 |
| FedFN - ResNet | 80.81 | 82.30 | 84.19 |

Table 14: Results with multiple runs for CIFAR-10 dataset in one class settings with 10 clients for both CNN and ResNet models.

| Methods | CNN | ResNet |
|---|---|---|
| FedAvg | $55.76_{\pm 0.55}$ | $55.34_{\pm 0.78}$ |
| FedYogi | $73.76_{\pm 0.70}$ | $76.90_{\pm 2.80}$ |
| FedFN | $76.85_{\pm 0.37}$ | $80.55_{\pm 0.17}$ |
| FedLN | $\mathbf{77.21}_{\pm 0.85}$ | $\mathbf{83.40}_{\pm 4.90}$ |

**Modifications of ResNet.** Last but not least, we test different modifications of ResNet in Table 13. We denote $n_{MV}$ inside as the following block:

$$\mathsf{block}(\boldsymbol{x}) = \rho(\boldsymbol{x} + \boldsymbol{A}_2 \circ n_{MV} \circ \rho \circ \boldsymbol{A}_1(\boldsymbol{x})),$$

and by n inside we use:

$$\mathsf{block}(\boldsymbol{x}) = \rho(\boldsymbol{x} + \boldsymbol{A}_2 \circ n \circ \rho \circ \boldsymbol{A}_1(\boldsymbol{x})),$$

for each block. Note that we always add the last-layer feature normalization for these two methods. We can see that both $n_{MV}$ and n help the improvement of FedFN, and they narrow the gap between FN and LN.

**Multiple seeds.** In Table 1, we reported the last-iteration accuracy for a single run. In Table 14, we take three runs for CIFAR-10 with one class settings for both CNN and ResNet models and report the mean and standard deviation for the three runs. Our results show that the comparison is consistent across different runs. Due to computation constraints we could not afford to run multiple seeds for all our experiments.

# F   Additional Related Work

**Normalization** is not a new idea, and it can be at least traced back to whitening. As suggested by Le-Cun et al. (1998), we should shift the inputs so that the average is nearly zero and scale them to have the same covariance. Simoncelli & Olshausen (2001) also explained whitening in computer vision through principal components analysis. Normalization is also a standard technique in training neural networks if one treats the inputs as the intermediate layers. Depending on which dimensions to normalize, it could be batch normalization (Ioffe & Szegedy, 2015), weight normalization (Salimans & Kingma, 2016), instance normalization (Ulyanov et al., 2016), layer normalization (Ba et al., 2016), and group normalization (Wu & He, 2018). Specifically, layer normalization is widely used in modern neural architectures such as Transformers (Vaswani et al., 2017). More recently, Zhang & Sennrich (2019) proposed Root Mean Square Layer Normalization (RMSNorm) which dispenses with the mean shift in layer normalization. This shares some similarities with our feature normalization. However, their motivation is quite different from ours.

There has been recent work that aims to understand normalization methods. For example, Santurkar et al. (2018) explained that batch normalization can help optimization; Dukler et al. (2020) provided convergence results for two-layer ReLU networks with weight normalization with NTK analysis. Lately, Lyu et al. (2022) studied the convergence of gradient descent with weight decay for scale-invariant loss functions. None of the previous work provided an analysis of the expressive power and local overfitting with softmax and cross-entropy loss.

**Expressive power.** The expressive power of (normalized) neural networks has been studied in Raghu et al. (2017) and Giannou et al. (2023). In Raghu et al. (2017), the authors proposed a new measures of expressivity and trajectory regularization that achieves similar advantages as batch norm. In Giannou et al. (2023), the authors argued that by tuning only the normalization layers of a neural network, we can recover any fully-connected neural network functions. Neither of them provides the same result as our Proposition 3.

