# OpenReview forum: "Understanding the Role of Layer Normalization in Label-Skewed Federated Learning"
_TMLR — Accepted by TMLR_

### Review · Reviewer_vPYs · 2023-11-10

**Summary Of Contributions:**

The paper investigates effectiveness of layer normalization (LN) in federated learning (FL) with non-i.i.d. data. While LN has proven effective in this context, the underlying reasons for its success have not been well understood. The authors establish a connection between layer normalization and the label shift problem in federated learning. They find that, as the distribution of labels among clients becomes more imbalanced, layer normalization becomes increasingly beneficial. This happens because LN controls local overfitting by mitigating feature collapse. Further, they introduce the concept of feature normalization (FN), a mechanism within LN that applies normalization to latent feature representations before the classifier head.  LN and FN control feature collapse and local overfitting on heavily skewed datasets, leading to accelerated global training.  The empirical findings demonstrate that normalization, particularly in the form of FN, significantly improves performance on standard benchmarks, particularly under extreme label shift conditions. Their results confirm that FN is a vital component of LN, enhancing convergence in FL and maintaining robustness to learning rate variations, especially in scenarios with extreme label shift.

**Audience:**

Yes

**Claims And Evidence:**

Yes

**Requested Changes:**

Please address the comments above in the weaknesses section.

**Strengths And Weaknesses:**

Strengths

1. The paper is well written and easy to follow and self-contained with sufficient background on normalization and federated learning.

2. It presents a study of layer normalization (LN) in federated learning(FL) setting and provides insights into when and why LN helps in FL. The authors observe that when the distribution of labels is uneven among clients, layer normalization becomes more helpful. This is because layer normalization prevents the model from feature collapse, which helps control overfitting and avoids losing important features.

3. They introduce the idea of feature normalization (FN), a process within LN that normalizes latent feature representations before the classifier head. While this doesn't increase the model's expressive capabilities, LN and FN effectively manage feature collapse and local overfitting in datasets with significant imbalances, which speeds up global training.


Weaknesses:


1. It is not quite clear what the empirical results are reporting? Is it the mean of the accuracy over multiple runs with different random seeds? Could you also plot/report the std. deviations?


2. The comparison in most figures is between FedAvg and variations with different normalizations, however table 1 suggests FedYogi is pretty close to FedLN, FedFN.

3. Minor comments:

   a. Please organize the tables, figures in pages 8-11 to better utilize the space.

   b. I personally do not like “All you need”  titles and the paper is full of this phrase. I would suggest using a shorter and more crisp title that matches with the paper’s findings.

---

> ### Author Response · Authors · 2023-12-01
> **Response to Reviewer vPYs**
>
> Thank you for your feedback and acknowledgment of our contribution. We address your concerns as follows:
>
> **Q1:** It is not quite clear what the empirical results are reporting? Is it the mean of the accuracy over multiple runs with different random seeds? Could you also plot/report the std. deviations?
>
> **A1:** For each setting, we report the last-iterate accuracy of a single run, which consists of 10,000 global rounds. We have updated Table 1 with this information and the extended description of our experimental setup in Appendix C.3.
>
> Due to limited computational resources, we are unable to run the full set of experiments with multiple seeds at this point. Nevertheless, we present some preliminary trials of three runs for CIFAR-10 with one class setup for both CNN and ResNet models. Our results show that the comparison is consistent across different runs. These results can also be found in Table 14.
>
>
> | Methods | CNN |  ResNet |
> | -------- | -------- | -------- |
> | FedAvg    | 55.76 ± 0.55     | 55.34 ± 0.78     |
> | FedYogi   | 73.76 ± 0.70     |76.90 ± 2.80        |
> | FedFN     | 76.85 ± 0.37     | 80.55 ± 0.17     |
> | FedLN     | 77.21 ± 0.85     | 83.40 ± 4.90      |
>
>
>
> **Q2:** The comparison in most figures is between FedAvg and variations with different normalizations, however, Table 1 suggests FedYogi is pretty close to FedLN and FedFN.
>
> **A2:** If we look at Table 1, we can see that the gap between FedYogi and FedLN is still significant. For example, for CIFAR-10 we have a 6% &mdash;  10% improvement and for TinyImageNet we have a 12% &mdash; 15% improvement. We can treat Yogi and LN/FN as orthogonal approaches to accelerate the convergence and we can combine them. We have done the combination in Table 12. Also, we need to point out that FedYogi requires additional hyperparameter tuning for each dataset, and it does not always perform well, e.g. in the TinyImagenet dataset in Table 1.
>
>
> **Q3a:** Please organize the tables, and figures on pages 8-11 to better utilize the space.
>
> **A3a:** Thank you for the suggestion. We have reorganized all the tables in the draft to make sure they are near the corresponding mention in the main text.
>
> **Q3b:** I personally do not like “All you need” titles and the paper is full of this phrase. I would suggest using a shorter and more crisp title that matches with the paper’s findings.
>
> **A3b:** The major finding of this paper is that "Layer/Feature Normalization" is essential for label shift problems in federated learning, which is why we used "All you need," but you're right that this phrase may not fully reflect our findings. We have removed it from the title and the body and modified the title as well.

---

### Review · Reviewer_JL46 · 2023-11-16

**Summary Of Contributions:**

The paper attempts to explain the role of Layer Normalization in federated learning. The authors identify feature normalization (normalizing pre-classification layer feature embeddings) as a key element that improves the quality of learning and connects it to layer normalization. They have enough numerical examples that validate the ability of their method. They also provide a theoretical analysis based on layer normalization that adds to their empirical results.

**Audience:**

Yes

**Broader Impact Concerns:**

None.

**Claims And Evidence:**

Yes

**Requested Changes:**

See previous section.

**Strengths And Weaknesses:**

[S1] The paper is very well written, and the ideas are presented clearly. The concepts are well-connected and the paper is easy to follow.

[S2] The topic of the paper is relevant given the ubiquitous use of normalization techniques in practice.

[S3] The numerical examples are a good proof that the method works. As a comment, I would like to see experiments turning LN/FN on and off for each of the algorithms considered.

[W1] What is the role of FL in this paper? What would be different than a centralized case? I fail to understand why the authors decided to write this paper in the context of FL.

[W2] The fundamental finding of the paper -- the role of FN in LN -- seems to me as not convincing. That is to say, the difference between LN and FN is the quality is normalizing (or not) the last layer. Therefore, the role of FN is not properly explained or understood. Even more so, the theory of the paper is regarding the relationship between these two normalizations. The empirical results show that LN is better than FN, therefore why is FN relevant?

[W3] In Propositions 1 and 2, the bias term =0 seems like an extremely difficult assumption to observe in practice and therefore, the relevance of these two propositions is seriously compromised. The authors should provide a better explanation of why these two propositions are relevant findings.

[W4] The paper does not provide enough details on the number of agents. They are spread out in the appendix, but I would suggest adding this data to each individual table with results.

[W5] The paper does not fully mention the topic of privacy which is of vital importance in FL. To compute LN/FN you need data from batch sizes which requires clients to provide local and private data. How is privacy assured in LN?

[W6] The paper does not mention the problem of communication when addressing the normalizations. The central agent needs to combine the local information from all agents and therefore more needs to be communicated. How does this affect training time and communication costs?

[W7] Proposition 3 and its explanation need more details. The way it is currently presented, and the importance/significance of this result is not clear. Even more so, the statement of the proposition is not self-contained and so it is difficult to assess its relevance. After carefully reading it, and looking at the proof (which I believe it is correct) I do not understand the point of the proposition, what message is it trying to convey? Also, adding to W1, the number of agents plays no role in it.

[W8] What is the relevance of Theorem 1 with respect to LN? There is little explanation provided and intuitively, if the norms diverge, the role of LN seems to be irrelevant. The authors should further clarify this point.

[W9] There are some important and relevant works that have not been included such as:

AN AGNOSTIC APPROACH TO FEDERATED LEARNING WITH CLASS IMBALANCE Shen et al.
A Theoretical Analysis of the Learning Dynamics under Class Imbalance Francazi et al.

---

> ### Author Response · Authors · 2023-12-01
> **Response to Reviewer JL46 (Part 1)**
>
> Thank you for your very careful review of our paper, and for the comments, corrections and suggestions that ensued.
>
> **Q0:** As a comment, I would like to see experiments turning LN/FN on and off for each of the algorithms considered.
>
> **A0:** We have the experiments with LN/FN on and off for FedAvg/FedYogi in Table 12, and the improvement of LN/FN is quite consistent. We are currently running it for other algorithms which may take some time.
>
> **Q1:** What is the role of FL in this paper? What would be different than a centralized case? I fail to understand why the authors decided to write this paper in the context of FL.
>
> **A1:** In this paper, first, we have shown the efficacy of layer normalization in label-skewed federated learning. After that, we aim to understand the effectiveness through both theoretical and empirical perspectives.
>
> There are a few major differences between the centralized case and our FL settings:
>
> * In our FL settings in Figure 4, FN is highly effective in accelerating the training process, but in the centralized case in Figure 6, FN converges similarly to the vanilla network. Figures 4 and 6 contrast the effect of normalization on convergence speed in the federated vs. centralized setting. The difference is most apparent for feature normalized networks: feature normalization leads to dramatic improvements in the federated learning setting (Figure 4, Table 1) whereas they make no difference in the centralized setting (Figure 6). This special relationship between normalization and federated merits further study, which is indeed the motivation of our investigation.
> * In our FL settings, each client has access to only a few number of classes. Therefore, during local training, local models diverge from the global model as shown in Sec 4.2. However, in the centralized setting, data from all classes is available.
> * Due to the label skewness in FL, the FL convergence is much slower. If we compare Table 4 and Figure 6, we can see that the gap between LN and vanilla networks is still large after 10,000 global rounds in the FL settings, but the gap diminishes after 40 epochs.
> * In our FL settings in Table 4, Batch Normalization is not helpful, but in the centralized case in Figure 6, BN works as well as LN.
>
> **Q2:** The fundamental finding of the paper -- the role of FN in LN -- seems to me as not convincing. That is to say, the difference between LN and FN is the quality is normalizing (or not) the last layer. Therefore, the role of FN is not properly explained or understood.
>
> **A2:** There are two main differences between LN and FN:
> * LN uses Mean-Variance (MV) normalization for each layer, while FN uses scale normalization for each layer.
> * Under our Assumption 1, FN reduces to scale normalizing the last layer, while LN cannot be reduced to the last layer only.
>
> In fact, both Sec 4 and Sec 5 aim to explain the role of FN in LN. In Sec 4, we explored FN and LN theoretically and found that FN is the essential mechanism inside LN that overcomes label shifts and addresses the feature collapse problem. In Sec 5, we conduct a comprehensive empirical analysis of the role of FN in LN, as well as the role of different normalization methods.
>
> **Q3:** Even more so, the theory of the paper is regarding the relationship between these two normalizations. The empirical results show that LN is better than FN, therefore why is FN relevant?
>
> **A3:** Understanding the relationship between LN and FN is one of our main goals in this work. Our work reiterates that FN is the essential mechanism behind LN that addresses feature collapse and label shift. Admittedly, LN is always better in our empirical settings, but we do not propose FN as an algorithm that we should use to address the label shift problem. FN is relevant because it summarizes the key mechanism of LN. It improves our understanding of LN, and in addition, our understanding of the label-skewed federated learning problem.
>
>
> **Q4:** In Propositions 1 and 2, the bias term = 0 seems like an extremely difficult assumption to observe in practice and therefore, the relevance of these two propositions is seriously compromised. The authors should provide a better explanation of why these two propositions are relevant findings.
>
>
> **A4:** We respectfully disagree with the claim that it is an extremely difficult assumption. As we see in the table below and the revised Table 3 (for more complete results), the performance with and without the bias term is negligible. Therefore, our propositions remain relevant.
>
> | Methods                 | with bias |  without bias   |
> | ----------------------- | -----------| --- |
> | FedAvg | 58.57  | 56.82 |
> | FedFN - $n^L$           |  77.70    |77.05  |
> | FedFN - $n1^{L−1}$      |    74.46          |77.14     |
> | FedLN - $n_{MV}^L$      |  77.75            |77.16     |
> | FedLN - $n_{MV}s^{L−1}$ |  76.89            | 77.71    |

---

> > ### Author Response · Authors · 2023-12-01
> > **Response to Reviewer JL46 (Part 2)**
> >
> > **Q5:** The paper does not provide enough details on the number of agents. They are spread out in the appendix, but I would suggest adding this data to each individual table with results.
> >
> > **A5:** Thank you for your suggestion. The number of clients was already summarized in Table 8 in the appendix. To make it clearer, we have added this data to each table (Tables 1, 3, 4). Overall, our setup includes 10 clients for the CIFAR-10 dataset, 50 and 20 clients for the CIFAR-100 dataset with $n$ class(es) and Dir(0.1), respectively, and 200 clients for TinyImageNet.
> >
> > **Q6:** The paper does not fully mention the topic of privacy which is of vital importance in FL. To compute LN/FN you need data from batch sizes which requires clients to provide local and private data. How is privacy assured in LN?
> >
> > **A6:** Unlike batch normalization which requires the record of the running statistics of data during training, layer normalization (LN) and feature normalization (FN) do not record statistics of batches. Instead, in LN and FN, the normalization is done independently for each sample both in training and inference. In addition, this means layer/feature normalization is compatible with differentially private gradient descent, the main tool that offers principled privacy guarantees to federated learning. In fact, group normalization (which includes layer and feature normalization as special cases) is an essential part of high-performing differentially private deep learning (https://arxiv.org/pdf/2204.13650.pdf). Therefore, adding FN and LN to an FL system is unlikely to degrade privacy. We added this discussion to the paper in Section 4.
> >
> > **Q7:** The paper does not mention the problem of communication when addressing the normalizations. The central agent needs to combine the local information from all agents and therefore more needs to be communicated. How does this affect training time and communication costs?
> >
> > **A7:** Our paper mainly investigates the architecture design of models for FL which suggests using layer normalization or feature normalization. These methods will not change the number of parameters of the network, and cause minor additional computation at each client. Therefore, vanilla FedAvg/FedFN/FedLN have the same communication cost in sending models to a server at *each communication round*. Further, as implied in Table 1, due to faster convergence, FedFN requires fewer communication rounds to reach a pre-defined accuracy, and therefore, FedFN has a better total communication cost. We have added these benefits of FN to the paper in Section 4.
> >
> > **Q8:** Proposition 3 and its explanation need more details. The way it is currently presented, and the importance/significance of this result is not clear. Even more so, the statement of the proposition is not self-contained and so it is difficult to assess its relevance. After carefully reading it, and looking at the proof (which I believe it is correct) I do not understand the point of the proposition, what message is it trying to convey? Also, adding to W1, the number of agents plays no role in it.
> >
> > **A8:** Thank you for pointing out this issue. In this work, we are trying to understand federated layer normalization in two ways: expressive power and training dynamics. There are two reasons why FedLN is more powerful:
> > * Layer normalization is more powerful in that it can express better functions to fit the data.
> > * Layer normalization allows the training process to be faster.
> >
> > Our Proposition 3 is important because it rules out the first choice. We have made this point clearer in the draft. You are right that the number of agents does not play a role here since it talks only about the model and it can be applied for both centralized or decentralized settings.
> >
> > **Q9:** What is the relevance of Theorem 1 with respect to LN? There is little explanation provided and intuitively, if the norms diverge, the role of LN seems to be irrelevant. The authors should further clarify this point.
> >
> > **A9:** Thanks for pointing out the clarity issue. We have added more discussion after Theorem 1 and Figure 3. Theorem 1 points out the importance of controlling the feature/class embedding norms. If we do not add any feature/layer normalization, minimizing the label-skewed local dataset could result in divergent norms. Combined with Figure 3, this theorem argues that controlling the divergent feature norms is important, and both LN/FN are doing that.
> >
> > **Q10:** There are some important and relevant works that have not been included such as:
> >
> > [1] Shen et al. AN AGNOSTIC APPROACH TO FEDERATED LEARNING WITH CLASS IMBALANCE
> > [2] Francazi et al. A Theoretical Analysis of the Learning Dynamics under Class Imbalance
> >
> > **A10:** Thank you for the references. We have added them in a revised version.

---

### Review · Reviewer_Zjcq · 2023-11-18

**Summary Of Contributions:**

This paper studies the role of Layer Normalization in Federated Learning in the presence of extreme label shift (e.g., when each client in the federated learning setup has data from exactly one of the classes being distinguished). I understood the paper as making two main contributions: (a) establishing and investigating the effectiveness of Layer Normalization for Federated Learning under extreme label shift, and (b) hypothesizing that the mechanism behind this effectiveness is "feature normalization."

**Audience:**

Yes

**Claims And Evidence:**

Yes

**Requested Changes:**

See weaknesses above

**Strengths And Weaknesses:**

Strengths:

- The paper is very well-motivated, picks an interesting problem, and is well-structured in the sense that I could understand what was happening without being an expert in Federated Learning.
- The results in the paper are tested on a wide suite of tasks and benchmarks, and are also comprehensive in answering when and why layer normalization works.
- The experimental procedure is thorough and well-documented.

Weaknesses/Questions:

1. The Theorem statements are a bit too informal for my liking (e.g., Propositions 1 and 2) --- for example, what is a "proper domain"?
2. The writing could be improved - although the structure of the paper is nice, there are some grammatical errors and awkward sentences that make the paper a bit hard to read at times.
3. I don't believe this will work, because of the authors' presented intuition, but it would be good to check that other methods for preventing overfitting (e.g., tuning learning rate, increasing weight decay) do not have the same effect as LN.
4. Although I understand what the authors mean here, I would recommend removing or revising the paragraph that follows Proposition 3, since it's a bit misleading. In particular, Proposition 3 shows that any function that an LN network can represent can also be represented by a non-LN network---the authors use this to argue that the benefit must be something during training. However, since *not* every NN can be represented as an LN, this claim is not true ---the proposition doesn't rule out model class restriction as the underlying mechanism behind LN's effectiveness.

---

> ### Author Response · Authors · 2023-12-01
> **Response to Reviewer Zjcq**
>
> Thank you for your acknowledgment of our work and providing constructive feedback.
>
> **Q1:** The Theorem statements are a bit too informal for my liking (e.g., Propositions 1 and 2) --- for example, what is a "proper domain"?
>
> **A1:** The formal statements are in Appendix B (Propositions 1' and 2'). We have added some explanations in the main text and modified the propositions. We did not include all the formal details to improve the readability of the propositions.
>
> **Q2:** The writing could be improved - although the structure of the paper is nice, there are some grammatical errors and awkward sentences that make the paper a bit hard to read at times.
>
> **A2:** We are sorry that you find grammatical errors and awkward sentences in our draft. Could you kindly point out the locations? Otherwise, it would be hard to keep track of them.
>
> **Q3:** I don't believe this will work, because of the authors' presented intuition, but it would be good to check that other methods for preventing overfitting (e.g., tuning learning rate, increasing weight decay) do not have the same effect as LN.
>
> **A3:** In this paper, we are addressing the *local overfitting* problem (the gap between accuracies of local validation sets and global test sets), which is not the same as conventional overfitting (train/test gap on the same distribution) in the centralized setting. Therefore, typical methods that address conventional overfitting do not apply here. For instance, we tested the baselines for different learning rates in Figure 7, and it showed that learning rate selection cannot prevent the overfitting as much as normalization. Further, even FedProx (which is a form of regularization akin to weight decay), does not perform well as shown in Table 1.
>
>
>
> **Q4:** Although I understand what the authors mean here, I would recommend removing or revising the paragraph that follows Proposition 3 since it's a bit misleading. In particular, Proposition 3 shows that any function that an LN network can represent can also be represented by a non-LN network---the authors use this to argue that the benefit must be something during training. However, since not every NN can be represented as an LN, this claim is not true ---the proposition doesn't rule out model class restriction as the underlying mechanism behind LN's effectiveness.
>
> **A4:** Thank you for providing suggestions to interpret Proposition 3, which shows that every LN network can be represented with a usual NN. This proposition rules out the possibility that LN's effectiveness is because it can express better functions to fit the data. We have revised the paragraph under Proposition 3 to avoid misleading the audience.
>
> It is true that LN's effectiveness could be attributed to the model class restriction, but FN is totally equivalent to a usual NN, and FN still significantly outperforms the latter in our experiments. This implies that model class restriction may not be an explanation.

---

### Author Response · Authors · 2023-12-01
**Thank you all**

We would like to thank the action editor and all the reviewers for taking care of our manuscript. We have been through the reviews carefully and modified the draft accordingly. In order for easy navigation, we have colored the changes in blue.

---

### Decision · Action_Editor_owkf · 2024-01-22

**Recommendation:** Accept with minor revision

**Comment:**

This paper studies the use of layer normalization for federated learning when there is label skew.  This is prompted by the recent empirical success of layer norm in federated learning in non iid settings.  The reviewers all voted to accept the paper (all "leaning accept").  They found the paper well motivated, relevant and found the empirical results convincing.  However, all the reviewers seemed to find issues with the clarity of the paper.  Therefore, I recommend accepting the paper, but I would like the authors to make some minor revisions for clarity, addressing the reviewer comments.  Also please address the comments regarding the motivation for and clarity of the theoretical contribution of the paper.

**Audience:**

Federated learning is certainly an area of interest for the community.  Layer normalization for federated learning may be a bit niche, but it is relevant for sure.

**Claims And Evidence:**

The reviewers all indicated that the claims are supported by evidence.  One reviewer found the empirical evidence clear and thorough and another found the numerical evidence to be a strength.  However, one reviewer felt that the theory wasn't really justified.